# Ash1 and Tup1 dependent repression of the *Saccharomyces cerevisiae HO* promoter requires activator-dependent nucleosome eviction

Emily J. Parnell[1], Timothy J. Parnell[2], Chao Yan[3,4,5¤], Lu Bai[3,4,5], David J. Stillman[1]*

1 Department of Pathology, University of Utah Health Sciences Center, Salt Lake City, Utah, United States of America, 2 Bioinformatics Shared Resource, Huntsman Cancer Institute, University of Utah, Salt Lake City, Utah, United States of America, 3 Center for Eukaryotic Gene Regulation, The Pennsylvania State University, University Park, Pennsylvania, United States of America, 4 Department of Biochemistry and Molecular Biology, The Pennsylvania State University, University Park, Pennsylvania, United States of America, 5 Department of Physics, The Pennsylvania State University, University Park, Pennsylvania, United States of America

¤ Current address: New York Genome Center, 101 Avenue of the Americas, New York city, New York, United States of America

* david.stillman@path.utah.edu

**Data Availability Statement:** All ChIP-Seq data are being held in the NCBI GEO database under the GSE158180 Accession Number. All other data are

## Abstract

Transcriptional regulation of the *Saccharomyces cerevisiae HO* gene is highly complex, requiring a balance of multiple activating and repressing factors to ensure that only a few transcripts are produced in mother cells within a narrow window of the cell cycle. Here, we show that the Ash1 repressor associates with two DNA sequences that are usually concealed within nucleosomes in the *HO* promoter and recruits the Tup1 corepressor and the Rpd3 histone deacetylase, both of which are required for full repression in daughters. Genome-wide ChIP identified greater than 200 additional sites of co-localization of these factors, primarily within large, intergenic regions from which they could regulate adjacent genes. Most Ash1 binding sites are in nucleosome depleted regions (NDRs), while a small number overlap nucleosomes, similar to *HO*. We demonstrate that Ash1 binding to the *HO* promoter does not occur in the absence of the Swi5 transcription factor, which recruits coactivators that evict nucleosomes, including the nucleosomes obscuring the Ash1 binding sites. In the absence of Swi5, artificial nucleosome depletion allowed Ash1 to bind, demonstrating that nucleosomes are inhibitory to Ash1 binding. The location of binding sites within nucleosomes may therefore be a mechanism for limiting repressive activity to periods of nucleosome eviction that are otherwise associated with activation of the promoter. Our results illustrate that activation and repression can be intricately connected, and events set in motion by an activator may also ensure the appropriate level of repression and reset the promoter for the next activation cycle.

within the manuscript and its Supporting Information files.

**Funding:** This work was supported by National Institutes of Health grants GM121858 and GM121079, awarded to L.B. and D.J.S., respectively. The funders had no role in study design, data collection and analysis, decision to publish, or preparation of the manuscript.

**Competing interests:** The authors have declared that no competing interests exist.

## Author summary

Nucleosomes inhibit both gene expression and DNA-binding by regulatory factors. Here we examine the role of nucleosomes in regulating the binding of repressive transcription factors to the complex promoter for the yeast *HO* gene. Ash1 is a sequence-specific DNA-binding protein, and we show that it recruits the Tup1 global repressive factor to the *HO* promoter. Using a method to determine where Ash1 and Tup1 are bound to DNA throughout the genome, we discovered that Tup1 is also present at most places where Ash1 binds. The majority of these sites are in "Nucleosome Depleted Regions," or NDRs, where the absence of chromatin makes factor binding easier. We discovered that the *HO* promoter is an exception, in that the two places where Ash1 binds overlap nucleosomes. Activation of the *HO* promoter is a complex, multi-step process, and we demonstrated that chromatin factors transiently evict these nucleosomes from the *HO* promoter during the cell cycle, allowing Ash1 to bind and recruit Tup1. Thus, activators must evict nucleosomes from the promoter to allow the repressive machinery to bind.

## Introduction

Chromatin is generally repressive to transcription, limiting access of regulatory factors and RNA polymerase to the DNA [1]. However, nucleosomes are dynamic structures that can be moved, loosened or evicted under certain conditions, allowing regulatory proteins to associate with their binding sites. Alteration of nucleosomes is accomplished by remodeling complexes that use the energy of ATP hydrolysis to slide or evict nucleosomes and by histone-modifying factors that change the state of histones and their interaction with DNA [2,3]. This ability to dynamically modify nucleosomes allows transcription to be a regulated process, in which factor binding sites are concealed by nucleosomes until an appropriate stimulus leads to their movement or eviction. The access of transcription factors to promoter sites is thus dictated in part by chromatin state, and is an important aspect of gene regulation.

The *Saccharomyces cerevisiae HO* gene is an important model for examining the interplay between transcription factors and chromatin. The *HO* promoter is highly regulated, with a complexity more similar to higher eukaryotic promoters than typical yeast promoters, but with an ease of genetic manipulation [4]. Nucleosome positions across the *HO* promoter are well-defined [5,6]. The process of *HO* activation involves progressive waves of nucleosome eviction across the promoter during the cell cycle, ultimately reaching the transcription start site and allowing for association of RNA polymerase [7,8]. Nucleosomes are then quickly redeposited to restrict *HO* expression to a narrow window within G1 of the cell cycle, with only a few transcripts produced per cell [7,9,10].

Expression of *HO* is also regulated to ensure the gene product is present in only one of two cells from each mitotic division. Yeast cells divide asymmetrically, giving rise to a large mother cell and a smaller daughter cell. The *HO* gene is expressed only in haploid mother cells and encodes a site-specific endonuclease that initiates mating type interconversion by cleaving the *MAT* locus [9,11,12]. The ability of the mother, but not the daughter, to alter its mating type allows mother and daughter cells to subsequently mate, forming a diploid to enhance survival.

The *HO* promoter is unusually long for a yeast promoter, with known transcription factor binding sites extending to nearly 2 kb upstream of the transcription start site [13–15] and the next upstream gene at -3000 bp. In addition, a long ncRNA that initiates at -2700 affects *HO* promoter memory under specific conditions [16]. Upstream Regulatory Sequences URS1 (-1900 to -1200) and URS2 (-900 to -200) contain binding sites for activating transcription

factors [14,17,18]. Promoter activation proceeds as an ordered recruitment of factors, initiated by entry of the Swi5 pioneer transcription factor into the nucleus during anaphase [7,19–21]. Swi5 associates with two nucleosome-depleted regions (NDRs) in URS1 at -1800 and -1300 and recruits three coactivator complexes: the SWI/SNF chromatin remodeler, the SAGA complex with the Gcn5 histone acetyltransferase, and Mediator [13–15,22–25]. The coactivators are interdependent upon one another for their association with the *HO* promoter and are responsible for chromatin changes that promote expression, most notably the removal of nucleosomes that initiates within URS1 and then spreads to URS2 [7,26]. Sites for the SCB binding factor (SBF) within URS2 are occluded by nucleosomes for most of the cell cycle, but become exposed as nucleosome eviction spreads toward the transcription start site [7,8]. SBF recruits the coactivator complexes to URS2, allowing further propagation of nucleosome eviction to the TATA box and subsequent association of RNA polymerase and initiation of transcription [7].

Many repressors and corepressors are also required for maintaining the appropriate level of *HO* expression. The activities of these proteins antagonize those of the coactivators, providing a balance that ensures the precise timing and level of *HO* promoter activity [27]. Genetic screens have identified subunits of two histone deacetylase complexes, Rpd3 and Hda1, as negative regulators of *HO* expression [27–31]. These complexes act in opposition to the histone acetyltransferase activity of Gcn5, making the nucleosomes more repressive to transcription. At least two DNA-binding proteins recruit the Rpd3 complex to the *HO* promoter. The first, Ash1, is a GATA-family zinc finger protein that accumulates predominantly in daughter cells and is the critical determinant of mother-specific *HO* expression [32–34]. A definitive binding site(s) for Ash1 has not been identified, but it has been suggested to bind to YTGAT motifs throughout the *HO* promoter [34]. The second protein, Ume6, was originally identified as a meiotic regulator, and represses transcription of many genes [35]. It binds to a single site within the *HO* promoter in a nucleosomal linker between URS1 and URS2 [27].

Other negative regulators identified in genetic screens for inappropriate transcriptional activation [27] may antagonize the SWI/SNF complex at the *HO* promoter. The Isw2 ATP-dependent chromatin remodeler promotes the movement of nucleosomes into NDRs and could play a role in opposing the nucleosomal eviction caused by SWI/SNF [36]. Ume6 is known to recruit both Rpd3 and Isw2 to promoters and could be doing so at *HO* [37,38]. The Tup1 corepressor protein was also identified as a negative regulator of *HO* expression activation [27]. Tup1, usually found in complex with Cyc8 in a 4:1 ratio, is recruited to many promoters in yeast by a variety of sequence-specific DNA-binding proteins, and has been suggested to reduce expression by masking the activation domain of its recruiting protein, inhibiting its interaction with SWI/SNF [39–42]. Tup1 also has genetic and biochemical interactions with the Rpd3 and Hda1 histone deacetylase complexes, providing another possible mechanism for it to balance the action of coactivators at the *HO* promoter [43–47]. The manner in which Tup1 is brought to the *HO* promoter is not clear, as there are no known sites for Tup1 recruiters.

In this report, we expand upon our knowledge of the Ash1 and Tup1 negative regulators and their relationship to chromatin, both at the *HO* promoter and genome-wide. We demonstrate that Tup1 is recruited to the *HO* promoter via the Ash1 DNA-binding protein. Ash1 is thus responsible for bringing both Tup1 and Rpd3 to the *HO* promoter, and recruitment of Tup1 is independent of the Rpd3 complex. ChIP experiments showed nearly identical binding profiles for Ash1 and Tup1 across the *HO* promoter, and nucleosomes conceal their sites of association for most of the cell cycle. We used ChIP-Seq to identify other Ash1, Tup1 and Rpd3 sites throughout the *S. cerevisiae* genome to determine whether Ash1 has similar properties within other promoters. We found the vast majority of Ash1 sites display colocalization

with both Tup1 and Rpd3. Sites of 3-way overlap are mostly within NDRs in intergenic segments of the genome. Ash1/Tup1 association with nucleosomal *HO* promoter DNA is therefore a notable exception, suggesting that chromatin changes at *HO* may be necessary for association of not only the SBF activating factor but also the Ash1/Tup1 repressing factors. We demonstrate that Ash1 and Tup1 bind to the *HO* promoter only after the Swi5 activator binds and initiates nucleosome eviction. Artificially decreasing nucleosome occupancy at *HO* allowed Ash1 binding in the absence of the activator, suggesting that the presence of nucleosomes impedes association of Ash1/Tup1 until the *HO* promoter activation cascade has begun.

## Results

### Tup1 association with the *HO* promoter requires the presence of Ash1

In a previous study, we performed a genetic screen to identify negative regulators of the *HO* promoter [27]. One of the mutants isolated in the screen was a hypomorphic allele of *TUP1*, *tup1(H575Y)*, suggesting that Tup1 may play a role in repressing *HO* transcription. To determine whether Tup1 associates with the *HO* promoter, we tagged endogenous Tup1 with a V5 epitope and performed ChIP analysis in asynchronous cells. Tup1-V5 bound to the *HO* promoter with a predominant peak centered at approximately -1200 relative to the *HO* ATG (Fig 1A, blue; "Downstream Site"). Substantial binding also extended upstream to approximately -2100 (Fig 1A; "Upstream Site"), suggesting there may be at least two sites of association. Tup1 is recruited to yeast promoters by a variety of DNA-binding transcription factors [39,40]. We therefore sought to determine which protein is responsible for Tup1 association with the *HO* promoter. Our prior studies on the Ash1 repressor had shown that Ash1 has a binding profile at *HO* similar to that of Tup1 (Fig 1A, red), suggesting the possibility that Ash1 could be responsible for Tup1 recruitment to *HO*.

ChIP analysis of Tup1 binding in wild type and in an *ash1* mutant confirmed our hypothesis that Ash1 is necessary for most of the Tup1 localization to the *HO* promoter. Binding was substantially reduced, though not completely eliminated, in the *ash1* mutant, both at the main peak (Downstream Site; Fig 1B) and further upstream (Upstream Site). The presence of residual Tup1 binding in the *ash1* mutant above a "No Tag" control (Fig 1C) suggests there may be another factor(s) that plays a lesser role in recruiting Tup1 to the *HO* promoter. This is consistent with observations at other genes, in which it is typical for multiple factors to contribute to Tup1 recruitment [42].

*HO* expression is cell-cycle regulated such that only a few transcripts are produced per cell cycle at the very end of G1 phase [9,10]. The ordered recruitment of transcription factors and coactivators required for *HO* activation has previously been examined extensively by ChIP analysis in cells with a *GAL::CDC20* allele that can be arrested at G2/M and then released to allow synchronous progression through the cell cycle [7,21,48]. Three repressive DNA-binding factors, Ash1, Dot6, and Ume6, bind to the promoter after initial association of the Swi5 transcription factor but before *HO* expression [27]. We examined Tup1 binding using *GAL:: CDC20* synchronization and found that, as expected, Tup1 associated with the *HO* promoter at the same time as Ash1, 25 min after the cells were released from the G2/M arrest (Fig 1D). Binding of Tup1 throughout the time course was vastly reduced in an *ash1* mutant, measured at both binding locations within the promoter (Figs 1D and S1).

To further confirm the role of Ash1 in recruitment of Tup1 to the *HO* promoter, we overexpressed *ASH1* from a multicopy YEp plasmid and examined Tup1 binding in cells transformed with either an empty YEp vector or with YEp-*ASH1*. Overexpression of the *ASH1* gene was confirmed by RT-qPCR analysis (S2A Fig), and ChIP analysis showed elevated Tup1 binding to the *HO* promoter (Fig 1E). Concomitant with the recruitment of additional Tup1, *ASH1*

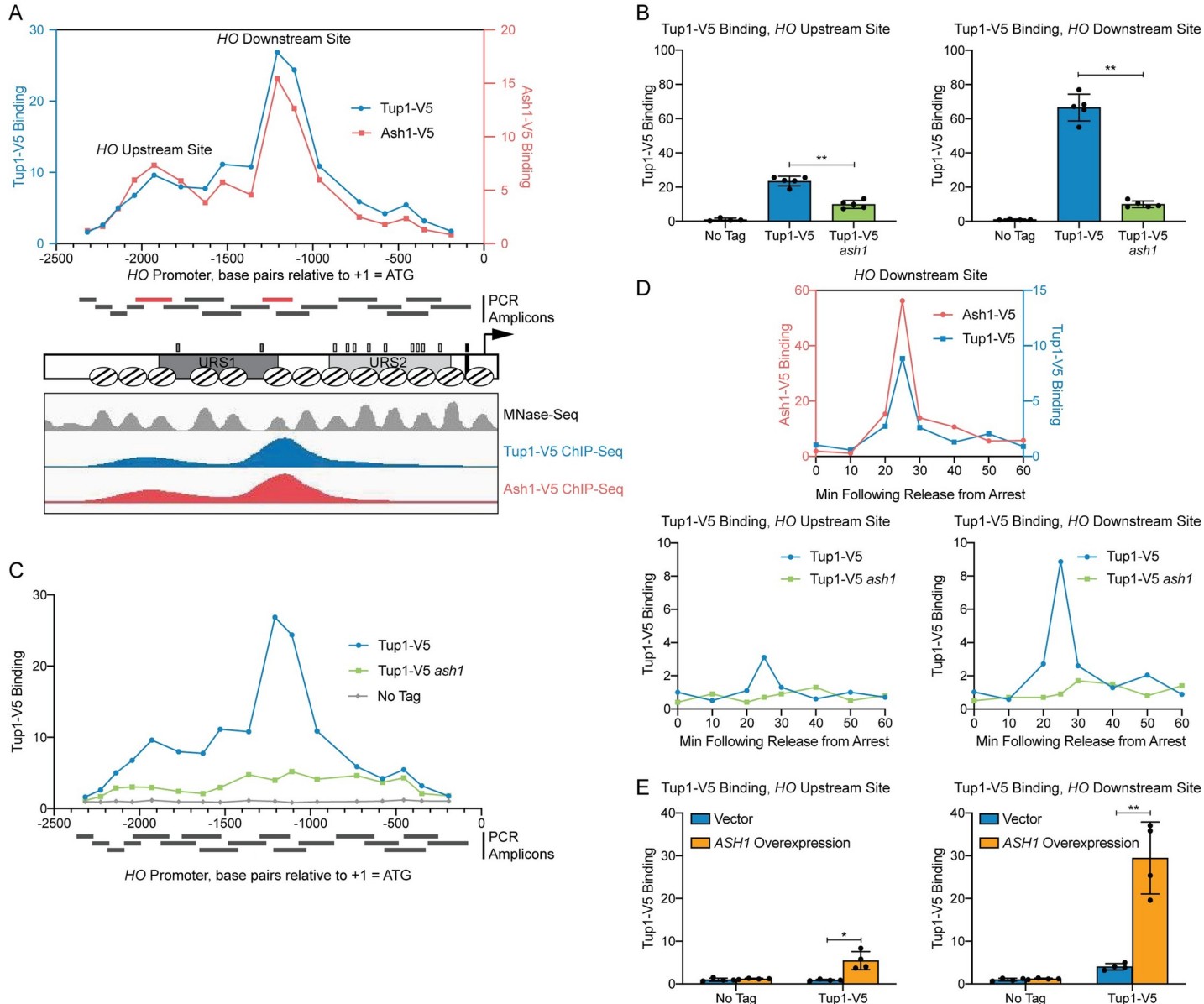

**Fig 1. Tup1 associates with the *HO* promoter via Ash1.** (A) There are two peaks of binding at the *HO* promoter for both Tup1 and Ash1. Binding of Tup1-V5 (blue; left y-axis) and Ash1-V5 (red; right y-axis) to the *HO* promoter was determined by ChIP, followed by qPCR with primers that span from -2300 to -200 in 75 to 150-bp intervals. Enrichment for each sample at *HO* was normalized to enrichment at an intergenic region on chromosome V (IGR-V) and to the corresponding input sample. Positions of the PCR amplicons are indicated with gray bars. Points on the graph correspond to the midpoints of these amplicons, with the x-axis indicating position across the *HO* promoter. Amplicons shown in red display the highest levels of binding of Tup1 and Ash1, labeled as "Upstream Site" (-2033 to -1823) and "Downstream Site" (-1295 to -1121). A schematic of the *HO* promoter shows the positions of nucleosomes from MNase-Seq [5] as ovals with slanted lines. The positions of Swi5 binding sites (dark gray small rectangles; within URS1), SBF binding sites (light gray small rectangles; within URS2), and the TATA element (black small rectangle) are also indicated. ChIP-Seq for Tup1-V5 (blue) and Ash1-V5 (red) shown in the bottom panel displays peaks of binding at the same Upstream and Downstream Site locations as the traditional ChIP in the top graph. Nucleosome sequences from MNaseq-Seq were trimmed to show only the central 76-nt, allowing their positions to be more easily viewed in the genome browser [5]. (B) Tup1 binding to the *HO* promoter is reduced in an *ash1* mutant at both the Upstream and Downstream sites. Tup1-V5 ChIP analysis at the *HO* promoter, showing enrichment at the Upstream Site (left; -2033 to -1823) and Downstream Site (right; -1295 to -1121). For each sample, binding at each *HO* site was normalized to its corresponding input DNA and to a No Tag control. Each dot represents a single data point, and error bars reflect the standard deviation. ** $p < 0.01$, * $p < 0.05$. (C) Tup1 binding to the *HO* promoter is not eliminated in an *ash1* mutant. Single samples of Tup1-V5 ChIP from Tup1-V5 (blue), Tup1-V5 *ash1* (green) and No Tag control (gray) strains were chosen from B and used for qPCR with primers that span the *HO* promoter, as in A. Enrichment for each sample at *HO* was normalized to enrichment at an intergenic region on chromosome V (IGR-V) and to the corresponding input sample. (D) Tup1 and Ash1 bind to the *HO* promoter at the same time in the cell cycle. Binding of Tup1-V5 and Ash1-V5 was measured by ChIP analysis with cells containing the *GALp::CDC20* allele and synchronized by galactose withdrawal and readdition. The 0 min time point represents the G2/M arrest, before release with galactose addition. Cells were harvested at the indicated time points following release (x-axis), and samples were processed for ChIP analysis. The top graph shows the coincidence of the timing

for binding of Ash1-V5 [red; left y-axis; 27] and Tup1-V5 (blue; right y-axis). Bottom graphs show binding of Tup1-V5 in wild type (blue) and *ash1* (green) backgrounds, at the *HO* Upstream Site (left) and *HO* Downstream Site (right). A single experiment is shown for simplicity; triplicate Tup1-V5 ChIP time courses are shown in S1 Fig. Enrichment for each sample at *HO* was normalized to enrichment at an intergenic region on chromosome I (IGR-I) and to the corresponding input sample. (E) *ASH1* overexpression results in increased Tup1 recruitment. Tup1-V5 ChIP analysis at the *HO* promoter, Upstream Site (left) and Downstream Site (right), is shown under conditions in which *ASH1* is overexpressed. Strains were transformed with a pRS426 (YEp-*URA3*) vector, either empty (blue) or containing *ASH1* (green). Binding at the *HO* sites for each sample was normalized to its corresponding input DNA and a No Tag control. Each dot represents a single data point, and error bars reflect the standard deviation. ** $p < 0.01$, * $p < 0.05$.

overexpression diminished *HO* expression (S2B Fig). A previous study demonstrated that YEp-*ASH1* caused an 8-fold drop in a mating type switching bioassay in mother cells, which reflects *HO* expression [33].

## Ash1 is sufficient to recruit Tup1 to an exogenous location

We next sought to determine whether Ash1 could recruit Tup1 to an ectopic location outside of the *HO* promoter. For this experiment, we constructed a Tup1-V5 strain in which a LexA DNA-binding site was integrated upstream of the *HIS3* gene on chromosome XV (Fig 2A). We then integrated a LexA DNA-binding domain and a FLAG tag at the 3' end of the endogenous *ASH1* locus to create a fusion protein. Association of Ash1-LexA(DBD)-FLAG with the ectopic LexA binding site should increase Tup1-V5 recruitment to that site if Ash1 is sufficient to recruit Tup1 (Fig 2A, right).

Ash1-LexA(DBD)-FLAG bound to both the LexA site upstream of *HIS3* and to the positive control promoter, *CLN3* (Fig 2B). Tup1-V5 binding at the ectopic *HIS3* site was minimal in the strain with native *ASH1*, but increased substantially in the strain containing Ash1-LexA (DBD)-FLAG (Fig 2C). As a comparison, Tup1-V5 bound to *TEC1*, the positive control promoter for Tup1 recruitment, in both strains (Fig 2C). We conclude that Ash1 is sufficient to recruit Tup1 to a location distinct from the *HO* promoter.

## Recruitment of Tup1 to the *HO* promoter by Ash1 is independent of Rpd3 (L)

Ash1 has been shown previously to repress *HO* transcription by virtue of association with the Rpd3(L) complex [6]. Ash1 is a substoichiometric member of Rpd(L), associating with the complex for only a portion of the cell cycle [6,49]. The Tup1 corepressor also interacts with multiple histone deacetylases, including Rpd3 [43,46]. We therefore considered the possibility that Tup1 associates with the *HO* promoter through an interaction with Rpd3(L) rather than through direct association with Ash1.

To address the question of whether Rpd3(L) and Tup1 are recruited by Ash1 independently and/or function independently for *HO* repression, we examined *HO* RNA expression in *rpd3* and *tup1* mutants using two methods. In the first method, we measured *HO* RNA in a bulk population of asynchronous cells (Fig 3A). In the second method, *HO-GFP* RNA was quantitated using single-cell time-lapse fluorescence microscopy, allowing the additional analysis of *HO* expression in mother versus daughter cells [50] (Fig 3B). An *rpd3* null single mutant did not change expression of *HO* in the bulk population, but single-cell analysis demonstrated that *HO* was expressed in approximately 50% of the daughter cells. The reason for this difference is not known, but may result from measurement of processed RNA in the bulk population as opposed to newly formed transcripts in the single cell experiment. It is also possible that unknown mother/daughter differences decrease the half-life of the *HO* mRNA in daughters, resulting in less apparent contribution to the bulk RNA total.

Null alleles of *tup1* show delayed progression of cells through G1 and therefore are not useful for monitoring the effect on *HO* expression in late G1 [27]. For these analyses, we therefore

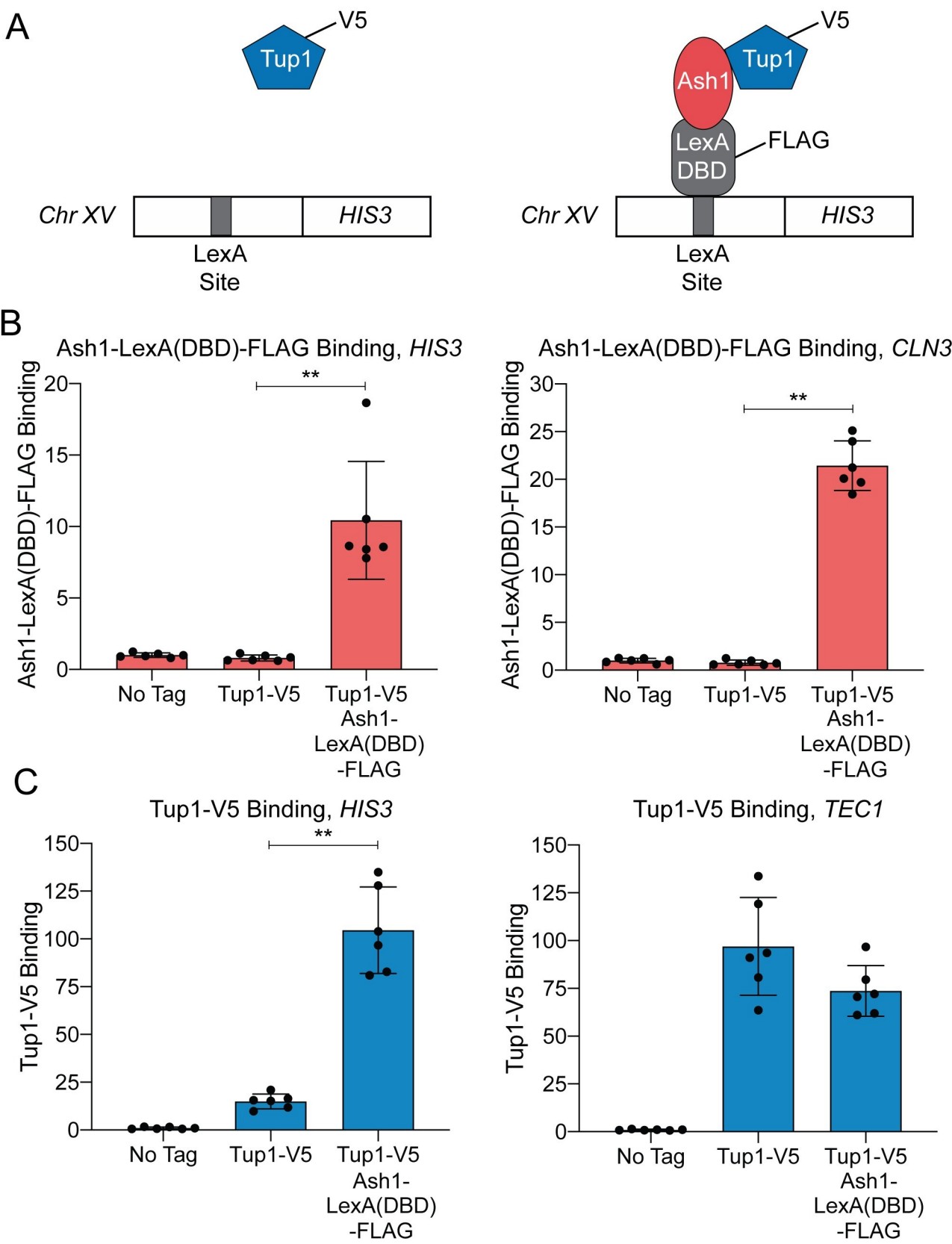

**Fig 2. Ash1-LexA(DBD)-FLAG recruits Tup1-V5 to a LexA binding site on chromosome XV.** (A) Schematic of experimental setup. Left–Strain with Tup1-V5 and LexA DNA-binding site integrated upstream of the *HIS3* gene on chromosome XV. Right–Strain with additional integration of Ash1-LexA(DBD)-FLAG. Recruitment of Tup1-V5 by Ash1-LexA(DBD)-FLAG brings Tup1-V5 to the LexA binding site on chromosome XV. (B) Ash1-LexA(DBD)-FLAG associates with the *HIS3* LexA site. ChIP analysis shows binding of Ash1-LexA(DBD)-FLAG to the LexA site upstream of *HIS3* (left) and to a positive control site at *CLN3* (right). Enrichment for each sample was normalized to its corresponding input DNA and a No Tag control. Each dot represents a single data point, and error bars reflect the standard deviation. ** $p < 0.01$. (C) Tup1-V5 is recruited to the *HIS3* LexA site in a strain with Ash1-LexA(DBD)-FLAG. ChIP analysis shows binding of Tup1-V5 to the LexA site upstream of *HIS3* (left) and to a positive control site at *TEC1* (right). Enrichment for each sample was normalized to its corresponding input DNA and a No Tag control. Each dot represents a single data point, and error bars reflect the standard deviation. ** $p < 0.01$.

used the *tup1(H575Y)* hypomorph that does not demonstrate a cell cycle delay. The *tup1 (H575Y)* single mutant showed a small increase in *HO* expression in both asynchronous cells (from 100% wild type to 120% *tup1(H575Y)*; Fig 3A) and in daughter cells in the single cell assay (from 2% wild type to 5% *tup1(H575Y)*; Fig 3B). In contrast to either single mutant, the double *rpd3 tup1(H575Y)* mutant had substantially increased *HO* expression in both assays, up to the level of an *ash1* mutant. In asynchronous cells, the level of expression in *rpd3 tup1 (H575Y)* and *ash1* was roughly 2-fold that of wild type, suggesting that daughter cells had fully gained the ability to express *HO*. This hypothesis was confirmed by the single-cell experiment, in which 96% of *rpd3 tup1(H575Y)* and 94% of *ash1* cells displayed daughter cell expression (compared to only 2% in wild type; see red in Fig 3B). The level of expression in daughter cells in the *rpd3 tup1(H575Y)* mutant was higher than in *ash1* cells (1.63 vs. 1.11), which may explain the slight increase in *HO* expression in the bulk population in the double mutant relative to *ash1*. This could occur due to off-target effects of the mutants that indirectly influence *HO* expression that are unrelated to their effects through Ash1.

The *HO* expression analyses demonstrate that mutation of both *rpd3* and *tup1* is required to achieve the increased *HO* expression in daughters that occurs in an *ash1* mutant, suggesting Ash1 could recruit the complexes independently. The mechanisms of repression by Rpd3(L) and Tup1 may be similar or distinct, yet the overall effect of combination of the two corepressors is severely diminished expression in daughter cells relative to mother cells. To more directly test the hypothesis that Tup1 is recruited to *HO* independently of Rpd3(L), we examined binding of Tup1-V5 in a *sin3* mutant. The subunits of the Rpd3(L) complex all interact with the Sin3 scaffold protein, and thus *sin3* mutants lack a structurally intact complex [51]. If Tup1 association with *HO* requires Rpd3(L) in addition to Ash1, then Tup1 should not be recruited to *HO* in the *sin3* mutant. We found that Tup1-V5 binding was similar in wild type and a *sin3* mutant (Fig 3C), demonstrating that Tup1 recruitment to the *HO* promoter is independent of the Rpd3(L) complex. Due to the cell cycle delay and severe flocculation phenotype of *tup1* null mutants, we were unable to accurately examine the reverse prediction, that Rpd3 recruitment is largely independent of Tup1. The *tup1(H575Y)* hypomorph still binds to the *HO* promoter, and thus is not ideal for testing this hypothesis. However, the increased *HO* expression in the *rpd3 tup1(H575Y)* double mutant relative to the *rpd3* single mutant suggests that both complexes are independently important for repression, and that if Rpd3 association does occur via Tup1, then Tup1 must have another activity that makes a separate contribution to repression.

## Ash1 is found at many genomic sites, where it colocalizes with Tup1 and Rpd3

More than a dozen DNA-binding transcription factors recruit Tup1 to promoters in yeast [39,40]. However, many sites of Tup1 localization are not bound by any known Tup1 recruiters [42]. This suggests there are other as yet unknown DNA-binding proteins that recruit Tup1, and Ash1 could be one of these factors. The only other known location of Ash1 binding

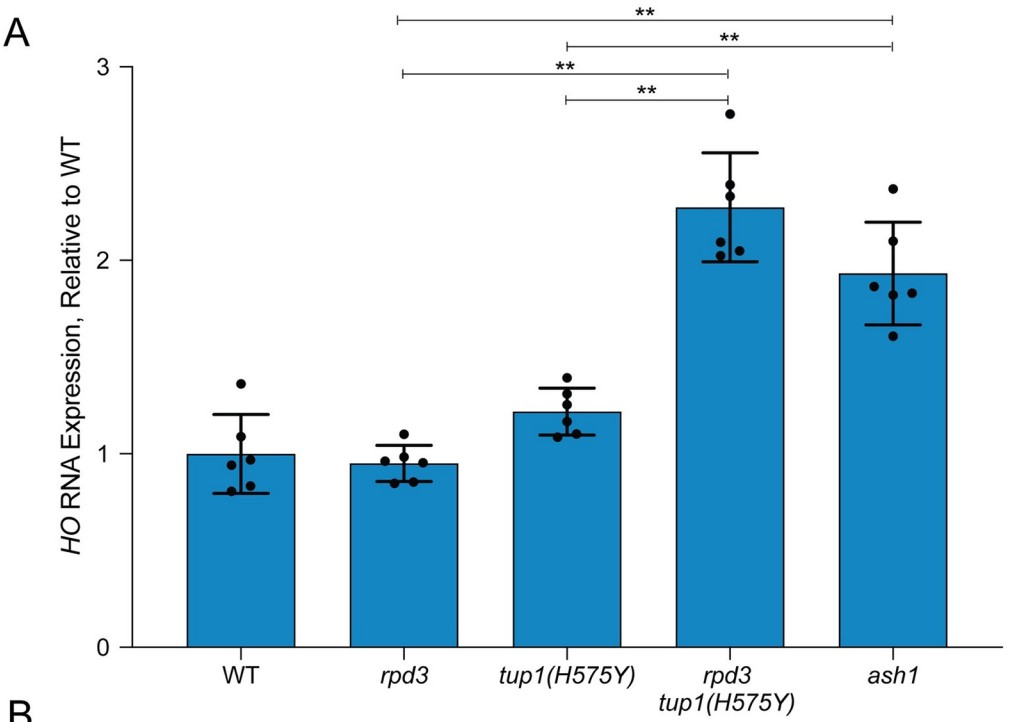

B

| | Mother | | | Daughter | | |
|---|---|---|---|---|---|---|
| **Strain** | **$P_{on}$ (%)** | **N** | **Level (on)** | **$P_{on}$ (%)** | **N** | **Level (on)** |
| WT | 98.0 ± 1.1 | 154 | 1.0 | 2.3 ± 1.3 | 131 | NA |
| *rpd3* | 99.1 ± 0.9 | 114 | 1.0 ± 0.03 | 53.8 ± 5.2 | 91 | 1.20 ± 0.06 |
| *tup1(H575Y)* | 95.3 ± 1.7 | 148 | 1.13 ± 0.03 | 4.5 ± 1.8 | 132 | 0.80 ± 0.22 |
| *rpd3  tup1(H575Y)* | 100 | 169 | 1.33 ± 0.03 | 95.7 ± 1.7 | 141 | 1.63 ± 0.04 |
| *ash1* | 98.4 ± 1.1 | 127 | 1.0 ± 0.02 | 93.5 ± 3.1 | 62 | 1.11 ± 0.06 |

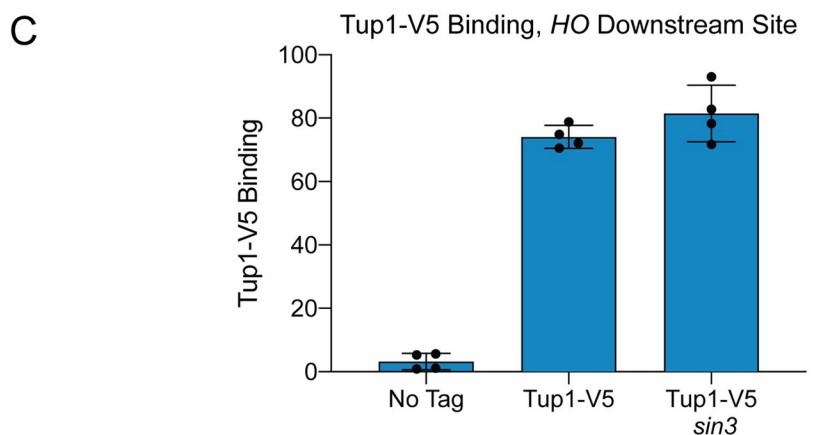

**Fig 3. Repression of *HO* transcription via Ash1 requires both Tup1 and Rpd3.** (A) RNA analysis shows that *tup1* and *rpd3* mutations are additive. *HO* mRNA levels were measured by RT-qPCR, normalized to *RPR1*, and expressed relative to wild type. Each dot represents a single data point, and error bars reflect the standard deviation. ** $p < 0.01$, * $p < 0.05$. (B) Single cell analysis shows that *tup1* and *rpd3* mutations are additive. Single cell *HO-GFP* fluorescence results for mother and daughter cells are shown, indicating the percentage of cells in which *HO-GFP* was on ($P_{on}$), the number of cells counted (N), and the relative levels of expression (Level–on), which were normalized to the wild type average, set at 1. Data for wild type, *rpd3* and *ash1* strains are from Zhang et al. [50]. (C) Tup1 recruitment is not affected by a *sin3* mutation. Binding of Tup1-V5 to the *HO* Downstream Site (-1295 to -1121) was determined by ChIP analysis, with each sample normalized to its corresponding input DNA and a No Tag control. Each dot represents a single data point, and error bars reflect the standard deviation.

is the *CLN3* promoter, where Ash1 cooperates with another daughter-specific factor, Ace2, to repress expression of *CLN3* in daughters [52,53]. To determine whether other sites of Ash1 binding exist, we performed ChIP-Seq with an Ash1-V5 strain. We also conducted parallel ChIP-Seq experiments with Tup1-V5 and Rpd3-V5 strains to assess how often Ash1 is present at sites that have both Rpd3 and Tup1 and whether there are subsets of promoters that are bound by Ash1/Tup1 or Ash1/Rpd3 pairs independently.

ChIP-Seq identified 250 peaks of Ash1 enrichment (Fig 4A and S3 Table), confirming our hypothesis that Ash1 binds to additional sites throughout the *S. cerevisiae* genome. This number is fewer than for either Tup1 (832) or Rpd3 (1377), which is not surprising since Tup1 and Rpd3 are more general factors that act at a larger number of genes, recruited by multiple different transcription factors, of which Ash1 is only one example. We confirmed the results of the ChIP-Seq by qPCR of ChIP eluate for each factor at specific target promoters, including several targets from different chromosomes with varying levels of enrichment (S4 Table). Values from qPCR correlated well with the ChIP-Seq values (S3 Fig).

The vast majority of Ash1 sites (99%) also displayed binding of either Tup1 or Rpd3 or both, demonstrating that the correspondence between Ash1 and these two repressive factors extends beyond the *HO* gene (Figs 4A and 4B and **S4**). Overlap of all three factors (Ash1, Tup1, Rpd3) was observed at 209 Ash1 peaks (84%; Fig 4A). A heat map of Ash1 peaks, displaying $\log_2$ fold enrichment of Ash1, Tup1 and Rpd3, shows varying levels of Tup1 and Rpd3 at different Ash1 locations (Fig 4C). Only a subset of the Tup1 and Rpd3 peaks overlap with those that are also bound by Ash1 (Fig 4A). Heat maps of Tup1 or Rpd3 peaks illustrate the substantial co-occupancy of these two factors, beyond the peaks that include Ash1 (S5 Fig; See also Figs 4B and S4 for genome snapshots). Of the three factors, Rpd3 had the largest number of peaks and therefore the greatest percentage of them that fail to overlap with the other two factors (Fig 4A). This was expected, based upon published studies of Rpd3 and the hypothesis that Rpd3 has a repressive role at specific promoters as well as a more general repressive function within open reading frames [54].

## Sites of co-occupancy of Ash1, Tup1 and Rpd3 are found within large intergenic regions

If Ash1 acts as a repressive transcription factor to recruit Tup1 and Rpd3 to locations other than *HO*, we would expect sites of Ash1, Tup1, and Rpd3 co-enrichment (ATR peaks) to be predominantly localized to intergenic regions, particularly those containing promoters, that would allow Ash1/Tup1/Rpd3 to regulate transcription of one or two genes from an upstream position. Consistent with this prediction, the majority of ATR peaks are positioned within intergenic regions (161 peaks, 77%; Tables 1 and S3). Additional peaks are located within either 5' or 3' UTRs (29 peaks; 14%). Only a very small number of ATR peaks have a summit within an ORF (6%) or over a ncRNA (<1%). Of the ATR peaks localized to intergenic regions, the vast majority (97%) are positioned in promoters, either unidirectional or bidirectional (Tables 2 and S3). Similarly, almost all ATR peaks within UTRs appear to be positioned

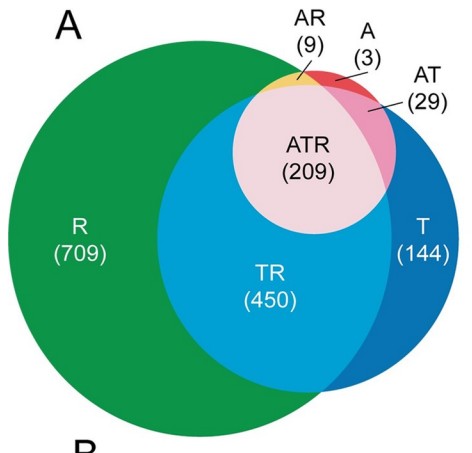

| Protein | ATR | AT | AR | TR | A | T | R | Total |
|---|---|---|---|---|---|---|---|---|
| Ash1 (A) | 209 (84%) | 29 (12%) | 9 (3%) | NA | 3 (1%) | NA | NA | 250 |
| Tup1 (T) | 209 (25%) | 29 (3%) | NA | 450 (54%) | NA | 144 (17%) | NA | 832 |
| Rpd3 (R) | 209 (15%) | NA | 9 (<1%) | 450 (33%) | NA | NA | 709 (51%) | 1377 |

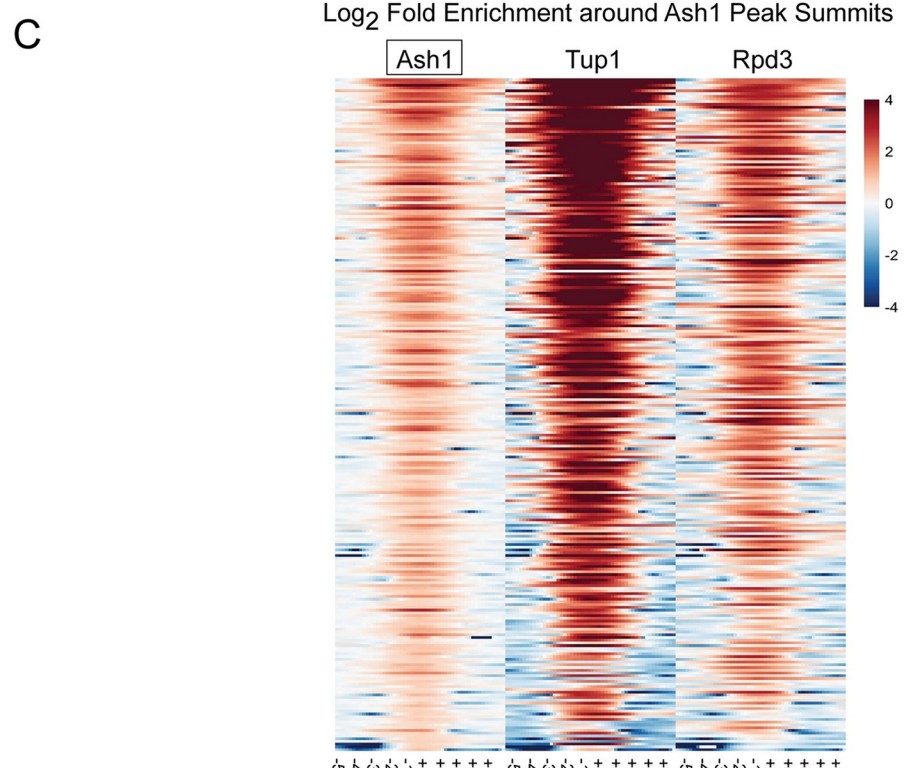

**Fig 4. Most Ash1 genomic sites are co-occupied by Tup1 and Rpd3.** (A) Sites of overlap between Ash1-V5, Tup1-V5, and Rpd3-V5 ChIP-Seq peaks were determined. The table displays the number of peaks and percentage of peaks in each category of single factor peaks and overlapping factor peaks, where A = Ash1, T = Tup1 and R = Rpd3. Overlap is shown visually in the Venn diagram at the left. (B) Snapshot of ChIP-Seq results from the Genome Browser IGV (Broad Institute), showing the sequenced fragment pileups for a portion of chromosome VII, with each factor autoscaled independently because each factor had a different ChIP efficiency. The top track (gray) shows MNase-Seq for nucleosome positioning reference [5]. The colored tracks show ChIP-Seq results for Ash1-V5 (red), Tup1-V5 (blue) and Rpd3-V5 (green). The bottom track displays gene annotations. Gene names are indicated only for those with start sites downstream of a site of Ash1-V5, Tup1-V5, and Rpd3-V5 (ATR) co-enrichment. Additional snapshots are shown in S4 Fig. (C) Heat maps depict the $\log_2$-fold enrichment of Ash1-V5, Tup1-V5 and Rpd3-V5 at Ash1-V5 peak summits genome-wide (246 peaks), displaying enrichment from -500 to +500 nucleotides relative to the center of each Ash1-V5 peak, in bins of 20-bp. The color scale at the right indicates the level of $\log_2$ fold enrichment for each factor. Each horizontal line depicts a single Ash1-V5 peak of enrichment. The number of peaks is less than that reported in Fig 4A because only the strongest peak per intergenic region was used for the analysis.

upstream to the neighboring gene's promoter rather than near its terminator. Only a few inter-genic and UTR peaks (5 total; 4 intergenic and 1 UTR) are located between convergent genes. Any potential role of these ATR sites in likely terminator regions is less clear.

Based on inspection of genome browser tracks, we noted that sites of ATR overlap appeared to occur in larger intergenic regions (Figs 4B and S4). We therefore compared the size distribution of all intergenic regions within the genome with those containing ATR peaks. The vast majority of yeast intergenic regions (close to 80%) are less than 500 nucleotides in length (S6 Fig), when considering transcriptional start and stop sites. In contrast, only 12% of those with ATR peaks are within this size range. Nearly 40% of ATR-containing intergenic regions are between 500 and 999 nucleotides, with the remaining approximately 50% greater than 1000 nucleotides in length (S6 Fig). Thus, ATR peaks are preferentially localized to larger promoter regions.

To determine the types of genes that could be regulated in part by Ash1 recruitment of Tup1 and Rpd3, we examined the functional nature of all ORFs downstream of intergenic ATR peaks. The largest group of possible ATR-regulated genes with a common feature is those encoding proteins located at the cell periphery, including structural components of the cell wall, proteins involved in budding, cell surface glycoproteins and membrane transporter proteins of many types (S5 Table). Several genes that control various aspects of the cell cycle are also down-stream of ATR peaks, including the G1 cyclins *CLN1*, *CLN2* and *CLN3*, and the B-type cyclins *CLB1* and *CLB2*. Genes involved in pseudohyphal growth, meiosis and sporulation were identi-fied, as well as genes encoding a variety of DNA-binding transcription factors. Some ATR peaks are located upstream of genes previously shown to be regulated by Tup1. Additional informa-tion on ORFs possibly regulated by ATR peaks can be found in S1 Appendix.

## Locations of Ash1, Tup1, and Rpd3 co-enrichment display differences in Ash1-dependence for Tup1 and Rpd3 recruitment

We assessed the contribution of Ash1 to Tup1 and Rpd3 recruitment at several genomic target sites to determine whether Ash1 is a predominant or minor recruiter at each location. Of the

**Table 1. Features of Ash1, Tup1, Rpd3 co-localized (ATR) peaks.**

| Location of Ash1 Peak[a] | Number of Peaks | Percent of Peaks |
|---|---|---|
| Intergenic | 161 | 77% |
| UTR | 29 | 14% |
| UTR/ORF Boundary | 5 | 2% |
| ORF | 13[b] | 6% |
| ncRNA | 1 | <1% |

a Determined by the position of the Ash1 peak summit.
b Eight of these are at the very 5' or 3' end of an ORF.

**Table 2. Relationship to Promoters of Ash1, Tup1, Rpd3 co-localized (ATR) peaks.**

| Promoter Direction | Number of Peaks[a] | Percent of Peaks |
|---|---|---|
| Single orientation | 69 | 46% |
| Divergent | 75 | 51% |
| Convergent | 4 | 3% |

a Only "Intergenic" peaks were used for analysis. Total number of peaks included is 148. Of the 161 peaks from Table 1, 13 were removed because one of the genes flanking the intergenic region was a tRNA or snRNA.

target sites we tested, *HO* displayed the greatest changes in Tup1 recruitment between wild type and an *ash1* mutant or *ASH1* overexpression (S4 Table). Sites upstream of other genes displayed moderate or small changes in Tup1 binding with alteration of *ASH1* levels. The relative level of Ash1 enrichment at each site did not predict the degree of change in Tup1 binding in the *ash1* mutant, and Tup1 binding was still detectable at all locations in the absence of Ash1. Most genes also showed a modest decrease in Rpd3 association upon removal of Ash1 (S4 Table). Similar to Tup1, Rpd3 binding was not eliminated. The most notable change in Rpd3 binding in an *ash1* mutant occurred at the *LTE1* gene, which is distinct from the other targets we examined because it is bound by Ash1 and Rpd3 but only weakly by Tup1. *LTE1* may represent a small class of genes in which Ash1 plays a more significant role in recruitment of the Rpd3 complex.

We also determined the level of Tup1 binding in a *sin3* mutant for this group of genes, to determine whether Tup1 association was dependent upon Rpd3 complex localization. Most did not show a substantial decrease in Tup1 binding in the *sin3* mutant, similar to *HO*, suggesting Rpd3 is not generally required for Tup1 recruitment (S4 Table). One exception is the *UBC4/TEC1* location, which does not have substantial binding of Ash1, but showed a decrease in association of Tup1 in the *sin3* mutant. This suggests there could be some locations with Tup1/Rpd3 dual association in which Rpd3 contributes to Tup1 recruitment.

## Ash1 and Tup1 associate with sequences encompassed within two nucleosomes of the *HO* promoter

In addition to identifying non-*HO* targets for Ash1, we planned to use the Ash1-V5 ChIP-Seq data to resolve some questions regarding the identity of Ash1 binding sites within the *HO* promoter. Our previous attempts to locate Ash1 binding sites based upon available data had been unsuccessful (See S2 Appendix for details). To identify an Ash1 binding motif from the genome-wide ChIP-Seq data, we used the central 100-bp surrounding the summit of the Ash1 peaks to search for motifs using the MEME-suite [55] and Homer [56]. The two most significant motifs identified by MEME are shown in S7 Fig. Motif 1 has low complexity, consisting largely of poly-A stretches, and was identified in 28% of the Ash1 peak sequences searched (S3 Table). This result is consistent with the presence of most Ash1 peaks within NDRs, which are frequently characterized by stretches of As and Ts [57]. Motif 2 resembles the binding site for Mcm1 [58,59] and was identified in 20% of the sequences (S3 Table). Mcm1 is an alpha helix transcription factor of the MADS box family that regulates expression of many genes, often in conjunction with interacting partner proteins at adjacent binding sites [60,61]. Sites for other transcription factors, such as Ume6, were identified in smaller subsets of peaks using Homer. No clear consensus motif emerged from either analysis or from additional searches using only ATR sites or Ash1 peaks within NDRs. We therefore suggest that Ash1 displays considerable flexibility in DNA recognition and/or that Ash1 binding to some locations is stimulated by interactions with other nearby DNA-binding factors (See S2 Appendix).

The possibility that Ash1 binds to a number of degenerate sequences suggests there may be multiple sites of Ash1 association at both the Upstream and Downstream Site locations of the *HO* promoter. These two peaks of Ash1/Tup1 binding coincide with the two nucleosomes of the *HO* promoter that flank the Swi5 binding sites (Nucleosome positions determined by MNase-Seq are shown in Figs 1A and 5; depicted by the yellow nucleosomes at -1890 and -1215 in Fig 5A); the MNase-Seq data are from log-phase cells, and may not reflect potential cell cycle changes or mother/daughter differences. To determine whether the sequence of these two nucleosomes contains most or all redundant sites of Ash1/Tup1 recruitment to the *HO* promoter, we replaced both nucleosome sequences, either singly or in combination, with the sequence of a positioned nucleosome from within the *CDC39* open reading frame. The sequence changes necessitated using different ChIP primers, indicated by the PCR amplicons upstream of the -1890 nucleosome and downstream of the -1215 nucleosome (Fig 5A). Replacement of the -1890 nucleosome slightly but significantly diminished binding of both Ash1 and Tup1 upstream of this nucleosome ("*HO* Left" Primers, Fig 5B and 5C) but not downstream of the -1215 nucleosome ("*HO* Right", Fig 5B and 5C). Likewise, replacement of the -1215 nucleosome dramatically decreased binding of both Ash1 and Tup1 downstream of this nucleosome but not upstream of the -1890 nucleosome. Thus, substitution of a single nucleosome affects Ash1/Tup1 ChIP levels in the vicinity, but does not affect Ash1 or Tup1 at the more distant relevant nucleosome. Substitution of both nucleosomes resulted in levels of Ash1/Tup1 binding at the "*HO* Right" location similar to replacement of the -1215 nucleosome alone (Fig 5B and 5C, Right). Double nucleosome replacement also diminished Ash1 binding at the "*HO* Left" location to a level similar to the single -1890 replacement, as expected (Fig 5B, Left). We did not observe the same effect for Tup1, because there was not an appreciable reduction in Tup1 binding at the "*HO* Left" location with substitution of both nucleosomes (Fig 5C, Left). This may be a consequence of substantially reduced binding of Tup1 at the Upstream Site relative to the Downstream Site (Fig 1B); the Upstream Site has a much smaller dynamic range, and it may be more difficult to detect slight differences in Tup1 binding due to sequence changes.

## Sequence replacement of two *HO* nucleosomes has a greater effect than an *ash1* mutation

As noted earlier and detailed in S2 Appendix, we mutated a variety of putative Ash1 binding site motifs but saw only modest effects on either Ash1 or Tup1 binding, or on expression of the *HO* gene. Significantly, replacement of the -1215 nucleosome had a greater effect on Ash1/Tup1 ChIP levels than any of the mutation combinations we had previously tested. However, the decreased dynamic range at the Upstream Site made it more difficult to determine the significance of the diminished binding due to replacement of the -1890 nucleosome. We therefore examined whether the changes in Ash1/Tup1 binding in the nucleosome replacement strains caused expected increases in *HO* expression, reasoning that if most or all Ash1 association sites were eliminated by the substitutions, *HO* expression should increase to the level observed in an *ash1* mutant.

Substitution of the -1890 nucleosome alone did not significantly affect *HO* expression (Fig 5D), which is consistent with the observation that the level of Ash1/Tup1 binding is much less at this nucleosome than at the -1215 nucleosome (Fig 1A). Substitution of the -1215 nucleosome did increase *HO* expression (Fig 5D), but the level of increase was much less than might be expected, given the substantial loss of Ash1/Tup1 association at the downstream site (Fig 5B and 5C, Right). However, substitution of both nucleosomes led to a more dramatic increase in *HO* expression, similar to an *ash1* mutant (Fig 5D). This level of *HO* expression was higher

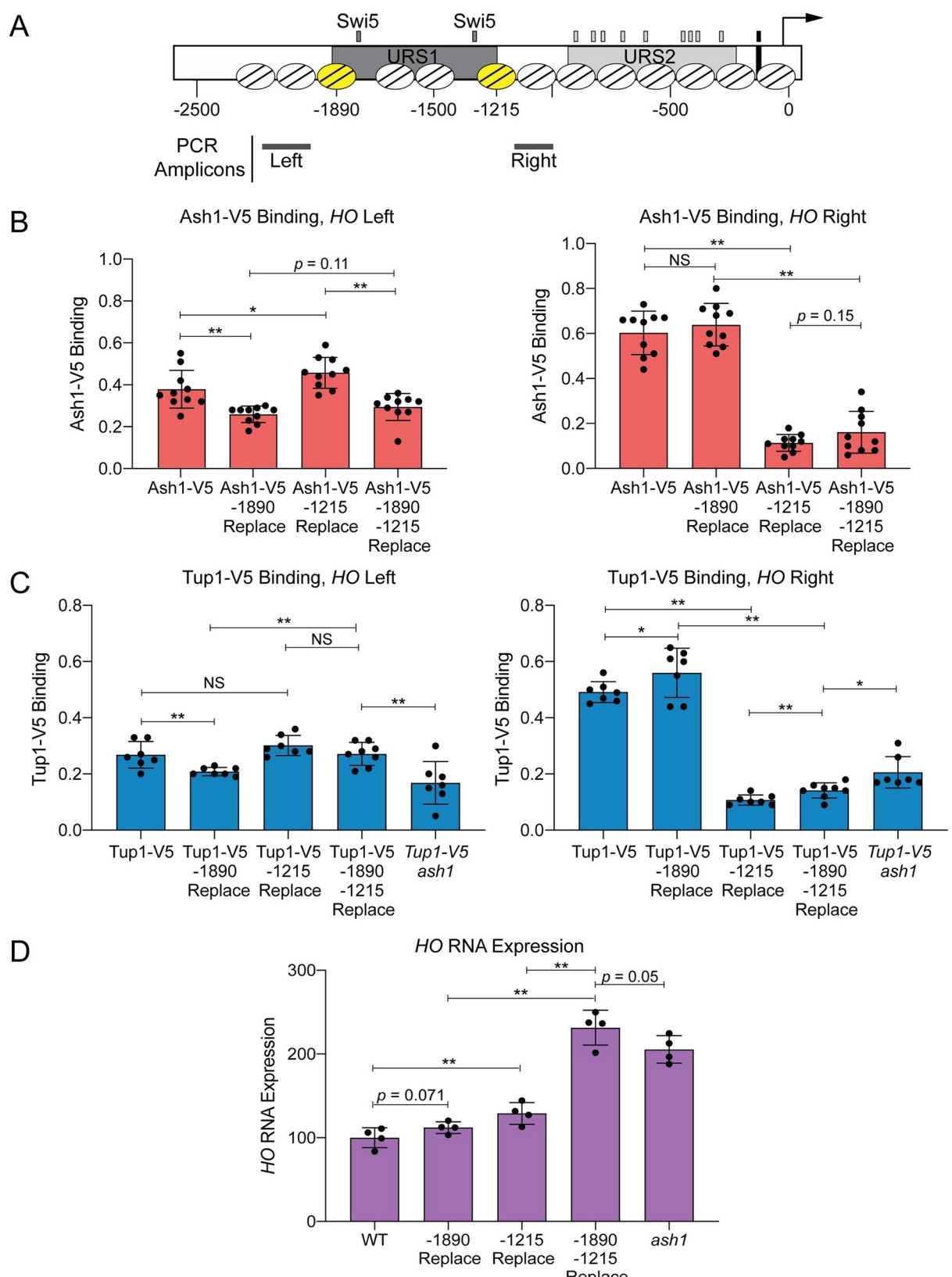

**Fig 5. Ash1 and Tup1 association with the *HO* promoter occurs within two nucleosomes that flank the NDRs containing Swi5 binding sites.** (A) A schematic of the *HO* promoter shows upstream regulatory sequences URS1 and URS2, Swi5 binding sites (dark gray small rectangles; within URS1), SBF binding sites (light gray small rectangles; within URS2), and the TATA element (black small rectangle). Positions of nucleosomes from MNase-Seq data [5] are shown as ovals with slanted lines. The two nucleosomes substituted with *CDC39* sequence (-1890 and -1215) are indicated in yellow. Positions of the Left and Right PCR amplicons are shown as gray bars. (B) Nucleosome substitutions reduce Ash1 binding. Ash1-V5 ChIP analysis at the *HO* promoter, showing enrichment upstream of the -1890 nucleosome ("-1890 Upper"; Left; -2195 to -1998) and downstream of the -1215 nucleosome ("-1215 Lower"; Right; -1137 to -978). "Replace" indicates that the sequence of the nucleosome listed (either -1890 or -1215) was substituted with the sequence of a nucleosome from the *CDC39* ORF. Binding at each *HO* site for each sample was normalized to *CLN3* as a positive reference control and its corresponding input DNA. Each dot represents a single data point, and error bars reflect the standard deviation. ** $p < 0.01$, * $p < 0.05$. (C) Nucleosome substitutions reduce Tup1 recruitment. Tup1-V5 ChIP analysis at the *HO* promoter, performed as in B, using *TEC1* as a positive reference control. (D) Substitutions at both nucleosomes increases *HO* expression to the level observed in an *ash1* mutant. *HO* mRNA levels were measured, normalized to *RPR1*, and expressed relative to wild type. Each dot represents a single data point, and error bars reflect the standard deviation. ** $p < 0.01$, * $p < 0.05$.

than in the -1215 substitution alone. Thus, Ash1/Tup1 binding was reduced most substantially by the -1215 substitution and more so by the double mutant, while *HO* expression was affected only partially by the -1215 substitution but very substantially by the double substitution. These results suggest that binding of Ash1 occurs predominantly within the sequence of the -1890 and -1215 nucleosomes and that the -1890 nucleosome is nearly as critical for *HO* regulation as the -1215 nucleosome, though the level of binding is much less.

## Sites of Ash1, Tup1, and Rpd3 co-occupancy are depleted for nucleosomes

The experiments above demonstrate that at the *HO* promoter, the majority of Ash1 and Tup1 binding occurs to sequences that appear to be within nucleosomes, determined by MNase mapping of nucleosome density in logarithmically growing cells [5]. Many transcription factors associate with sites that are in regions depleted of nucleosomes (Nucleosome Depleted Regions, NDRs) and the presence of nucleosomes generally inhibits binding of transcription factors [1]. To determine whether the Ash1/Tup1 binding at *HO* is unique or whether Ash1 is more likely to bind within sites of higher nucleosome density than other transcription factors, we compared the ChIP-Seq enrichment signals for Ash1-V5, Tup1-V5 and Rpd3-V5 with genome-wide MNase-Seq data [5]. Heat maps displaying the nucleosome density from -750 to +750 nucleotides relative to the summit of each Ash1 peak show that the central portion of the majority of Ash1 peaks lies within a region of low nucleosome density ([Fig 6A]). Peaks near the bottom of the heat map are more similar to *HO* in that they overlap with higher nucleosome densities. Like most transcription factors, Ash1 binding therefore largely occurs within NDRs, but a subset of locations has Ash1 association over nucleosomes, as measured in a bulk population of cells.

Similar plots for Tup1-V5 and Rpd3-V5 peaks demonstrate that each of these factors also has a group of peaks that overlap with NDRs, though the fraction of peaks with NDRs is less than for Ash1-V5 ([Fig 6B and 6C]). Of the three factors, Rpd3-V5 is the least likely to be recruited to sites within NDRs, consistent with the observations that Rpd3 has a more broadly repressive role and a known enzymatic function targeting nucleosomes [54]. As expected, plotting nucleosome density for only the ATR co-localized peaks shows a pattern similar to that for Ash1, with the majority of peaks overlapping regions of less nucleosome density ([Fig 6D]). Many of the Tup1 and Rpd3 peaks with low nucleosome density are thus sites of co-localization with Ash1. However, both factors clearly have additional binding locations within NDRs, consistent with the fact that both are recruited by transcription factors other than Ash1, which may also associate with sites of low nucleosome density.

To specifically identify ATR peaks other than *HO* that overlap with nucleosomes, we next categorized each intergenic ATR peak based upon the position of the Ash1 peak summit

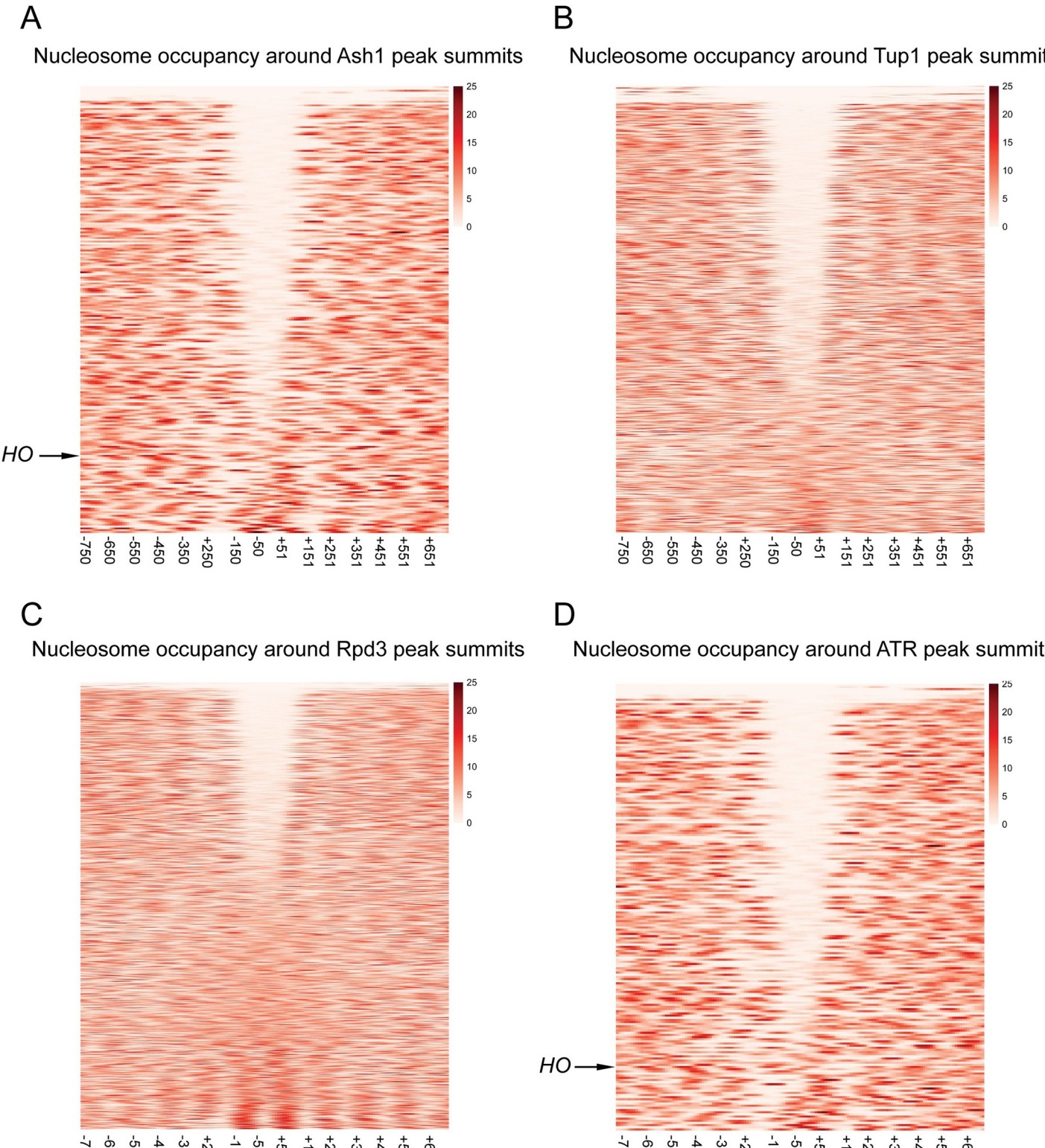

**Fig 6. Sites of Ash1, Tup1, and Rpd3 co-enrichment are found within nucleosome depleted regions (NDRs).** The MNase-Seq data was used to estimate the dyad position for each nucleosome, and this information was used to create a pileup of MNase-generated fragments, with each line representing a single gene. MNase-Seq heat maps are shown depicting the nucleosome occupancy surrounding the peaks for factor binding, displaying density from -750 to +750 nucleotides relative to the center of each peak, in bins of 20-bp. The color scale at the right indicates the level of nucleosome occupancy (fragments per million). (A) Each horizontal line depicts a

single Ash1-V5 peak, of 246 total peaks, with the *HO* peak indicated. The number of peaks is less than that reported in Fig 4A because only the strongest peak per intergenic region was used for the analysis. (B) Each horizontal line depicts a single Tup1-V5 peak, of 816 total peaks. (C) Each horizontal line depicts a single Rpd3-V5 peak, of 1343 total peaks. (D) Each horizontal line depicts a single peak of co-enrichment of Ash1-V5, Tup1-V5 and Rpd3-V5, of 209 total peaks, with the *HO* peak indicated.

relative to mapped NDRs and nucleosomes. Peaks were placed into one of three categories (Tables 3 and S3). "NDR" or "Nucleosome" peaks are those for which the summit of the Ash1 peak intersects with a mapped NDR or nucleosome, respectively. "Nucleosome/NDR Boundary" peaks are those located at the edge of a nucleosome or NDR, such that the summit of the peak lies within 25-bp of the edge of a mapped nucleosome. Some peaks were discarded from the analysis due to poorly-defined nucleosomes or insufficient MNase-Seq coverage from redundant sequence.

Three-quarters of the intergenic ATR peaks were positioned within NDRs (Table 3). The remainder were split between those that showed localization at an NDR/Nucleosome boundary and those positioned within nucleosomes. The type of ATR peaks similar to those at *HO* (S8A Fig) are thus in the minority, with only 13% of ATR intergenic peaks in which the sites of co-localization are found within mapped nucleosomes. Examples of each peak type are shown in S8B Fig.

### Association of Ash1 and Tup1 with the *HO* promoter requires the Swi5 activator and nucleosome eviction

Since a minority of Ash1 peaks are localized within nucleosomes, we considered the possibility that the Ash1 may not be physically able to bind to sequences within a nucleosome. Given that most Ash1 binding occurs within NDRs, a more likely scenario may be that at the "Nucleosome" sites, eviction would transiently reveal the Ash1 binding site, allowing Ash1 to bind and influence transcription.

We investigated this possibility using the *HO* gene, since previous studies have shown that *HO* promoter nucleosomes are evicted as the cell cycle progresses [7]. In cells synchronized by a *GAL::CDC20* arrest and release protocol, Ash1 binds to the *HO* promoter at 25 min after the release point [27] (Fig 7A). This occurs 5 min after the Swi5 transcription factor binds to the promoter (20 min following release) but before *HO* transcription occurs (starting at 30 minutes and peaking at 50 minutes following release; Fig 7A) [7,21]. Binding of Swi5 is the initial event that catalyzes a series of steps leading to activation of *HO* transcription. Swi5 recruits coactivators to the promoter, including the SWI/SNF chromatin remodeling complex, causing eviction of nucleosomes throughout and beyond URS1 [7,62]. The -1890 and -1215 nucleosomes containing Ash1 sites of association have already been evicted from URS1 at the 25 min time point when Ash1 binds [7,62]. Thus, it is likely that Ash1 is able bind to the *HO* promoter

**Table 3. Relationship of ATR Intergenic Peaks to Nucleosome Density.**

|  | Number of Peaks[a] | Percent of Peaks |
|---|---|---|
| Nucleosome Depleted Region (NDR) | 99 | 74% |
| Nucleosome / NDR Boundary [b] | 18 | 13% |
| Nucleosome | 17 | 13% |

a The 161 "Intergenic" ATR peaks from Table 1A were used for analysis. 24 ATR peaks could not be scored due to location within a region with poorly defined nucleosomes. An additional three were double peaks, in which only the larger of the two peaks was scored. The total shown here is 134.
b Peaks for which the Ash1 summit was within 25-bp of the edge of a mapped nucleosome.

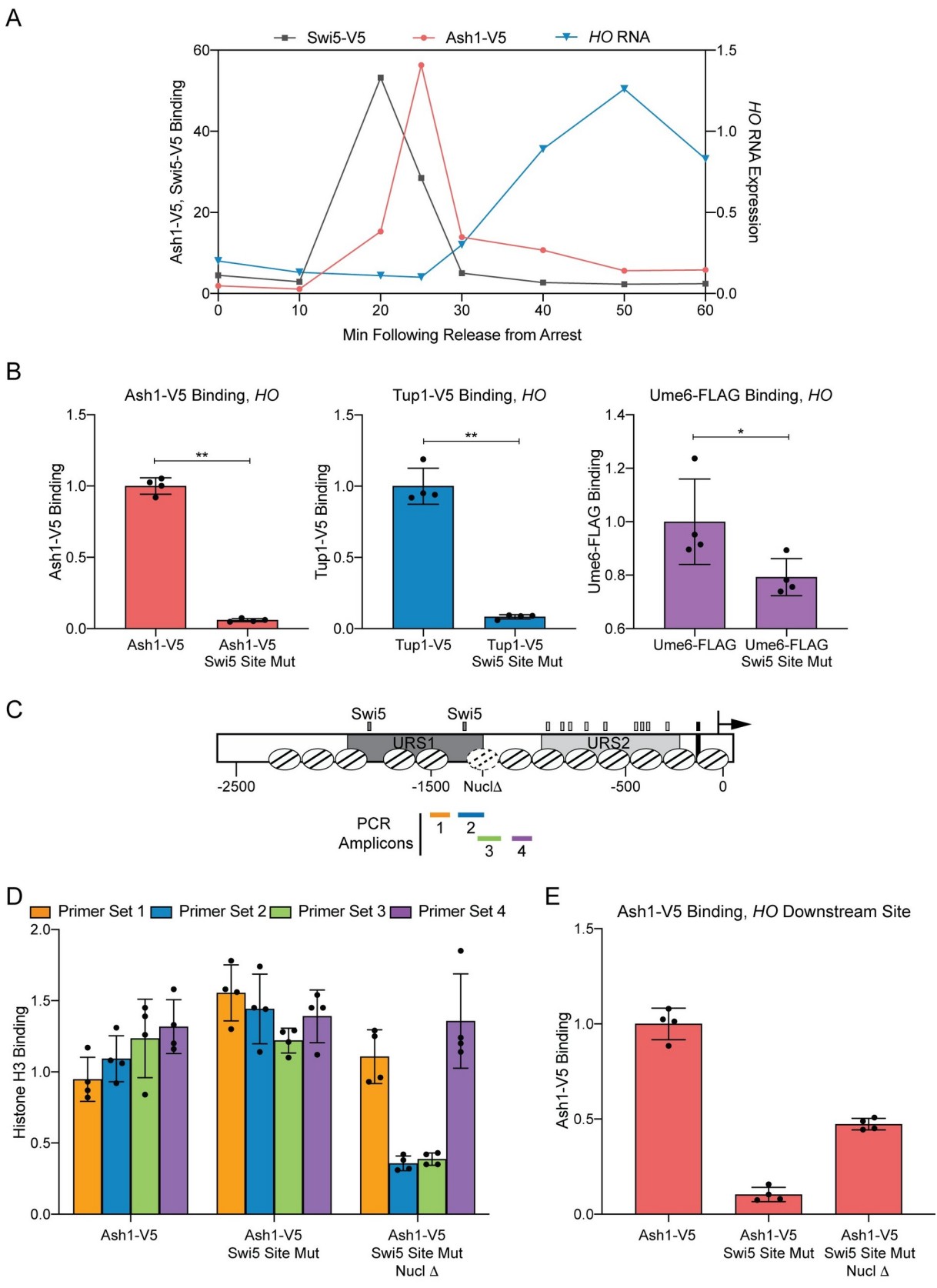

**Fig 7. Binding of the Ash1 repressor to the *HO* promoter only occurs under conditions of low nucleosome density.** (A) Cell cycle time course of Swi5 binding, followed by Ash1 recruitment, and finally, *HO* expression. ChIP and *HO* mRNA analysis were performed in Swi5-V5 or Ash1-V5 strains containing the *GALp::CDC20* allele and synchronized by galactose withdrawal and readdition. The 0 min time point represents the G2/M arrest, before release with galactose addition. Cells were harvested at the indicated time points following release (x-axis). Binding of Swi5 (gray; *HO* -1429 to -1158; left y-axis) and Ash1 (red; *HO* -1295 to -1121; left y-axis) was normalized to enrichment at an intergenic region on chromosome I (IGR-I) and to the corresponding input sample. *HO* mRNA levels (blue; right y-axis) were normalized to *RPR1*. (B) Swi5 binding is required for Ash1 binding and Tup1 recruitment. Ash1-V5, Tup1-V5 and Ume6-FLAG ChIP analysis, followed by qPCR with primers from *HO* -1295 to -1121. "Swi5 Site Mut" indicates strains in which both Swi5 binding sites are mutated and nonfunctional for *HO* activation. Binding at *HO* for each sample was normalized to its corresponding input DNA and to a positive reference control [*CLN3* for Ash1, *TEC1* for Tup1 and *INO1* for Ume6; 27]. Each dot represents a single data point, and error bars reflect the standard deviation. ** $p < 0.01$, * $p < 0.05$. (C) Schematic of the *HO* promoter with positions of nucleosomes from MNase-Seq shown as ovals with slanted lines. The "Nucl Δ" nucleosome with dotted lines indicates the -1215 nucleosome targeted for displacement by introduction of two Reb1 sites (TTACCC) that substitute for *HO* sequences from -1268 to -1262 and from -1194 to -1189. Positions of the PCR amplicons are indicated with colored bars. (D) H3 ChIP shows Reb1 sites lead to nucleosome loss. Graph shows histone H3 ChIP analysis using strains that are Ash1-V5 with Swi5 wild type binding sites (Ash1-V5) or Swi5 binding site mutations (Ash1-V5 Swi5 Site Mut) or Swi5 binding site mutations and nucleosomal substitutions with Reb1 sites to displace the nucleosome (Ash1-V5 Swi5 Site Mut Nucl Δ). qPCR was performed with ChIP material using the following primers: primer set 1 (orange) = *HO* -1497 to -1399; primer set 2 (green) = *HO* -1347 to -1248; primer set 3 (blue) = *HO* -1257 to -1158; primer set 4 (purple) = *HO* -1277 to -978. Binding at each *HO* site was normalized to an intergenic region on chromosome I (IGR-I) and to the corresponding input DNA and the No Tag control. ** $p < 0.01$, * $p < 0.05$. (E) Nucleosome loss partially restores Ash1 binding even in the absence of the normally required Swi5 activator. Ash1 binding was measured by ChIP, using the same chromatin samples as the histone H3 ChIP in D. Binding in each sample was measured by qPCR at *HO* -1295 to -1121 and normalized to the *CLN3* positive reference control and its corresponding input DNA. ** $p < 0.01$.

at this particular time because the nucleosomes covering its binding sites have been removed. If so, Ash1 may be similar to other transcription factors whose binding is restricted to NDRs.

If Ash1 requires nucleosome eviction at the *HO* promoter to promote binding, we expect that if we remove the capacity for nucleosome eviction, Ash1 should be incapable of binding. To examine this possibility, we constructed strains for measuring Ash1 binding in the absence of the Swi5 pioneer transcription factor. Without Swi5, there is no recruitment of SWI/SNF and no nucleosome eviction at the *HO* promoter [7]. We constructed strains with Swi5 binding site mutations a3 and b3 [63], which eliminate both Swi5 binding and *HO* expression, and assessed whether Ash1 and Tup1 could bind to the *HO* promoter in these conditions. ChIP assays showed that both proteins were virtually eliminated from the *HO* promoter in the strain with mutated Swi5 binding sites (Fig 7B). In contrast, the Ume6 repressive transcription factor, which associates with the *HO* promoter at a site that lies at least partially within a linker region [27], was not as strongly affected.

If Ash1 and Tup1 are unable to associate with the *HO* promoter in the absence of Swi5 because a nucleosome excludes them from binding, then experimental removal of the nucleosome should restore binding even in the presence of mutated Swi5 binding sites that prevent SWI/SNF recruitment. We therefore constructed a strain in which we introduced Reb1 binding sites within the -1215 nucleosome (Fig 7C; labeled "Nucl Δ"). Reb1 binding sites exclude the formation of nucleosomes [64,65]. We first performed histone H3 ChIP analysis to demonstrate that the Reb1 sites had changed the nucleosome density around the -1215 region. Primer sets 1 and 4, which lie outside of the -1215 nucleosome sequence, displayed either modest reduction (set 1, orange) or no change (set 4, purple) in H3 ChIP upon addition of the Reb1 binding sites (Fig 7D; compare "Swi5 Site Mut" to "Swi5 Site Mut Nucl Δ"). In contrast, primer sets 2 and 3, which overlap the -1215 nucleosome, showed dramatically decreased H3 ChIP enrichment when Reb1 binding sites were added (Fig 7D). Thus, the Reb1 binding sites were successful in reducing nucleosome occupancy over the nucleosome that contains the Ash1 downstream *HO* binding site(s).

We next measured Ash1 binding to these mutant promoters (Fig 7E). The Swi5 binding site mutations eliminated Ash1 binding, in agreement with the data in Fig 7B. Importantly, the reduction in nucleosome density caused by the Reb1 binding sites partially restored Ash1

binding, despite the absence of Swi5 and recruitment of the SWI/SNF remodeler. The Reb1 site eliminated the -1215 nucleosome, but the -1890 remained; synergy in binding between Ash1 at the -1890 and -1215 regions could provide a possible explanation for why the Reb1 site insertion only partially restored Ash1 binding. These experiments suggest that Ash1 binding to the *HO* promoter requires the nucleosomes covering its binding sites to be evicted, thereby exposing the binding sites. Thus, the *HO* promoter must undergo its initial activation steps in order for the Ash1 and Tup1 repressors to bind. This adds another level of complexity to our knowledge of *HO* promoter regulation and suggests an interplay between activation and repression factors is necessary for appropriate *HO* expression.

## Discussion

We have shown previously that the Tup1 corepressor functions as a negative regulator of *HO* expression, and here we demonstrate that Ash1 is the predominant recruiter of Tup1 to the *HO* promoter. ChIP-Seq revealed that Ash1 binds to many additional sites throughout the *S. cerevisiae* genome and colocalizes with Tup1 at 95% of these sites, most of which are also bound by Rpd3 (Fig 4A). Characterization of these sites provides insight into the genome-wide role of Ash1/Tup1/Rpd3 and aids in understanding the complexity and unique nature of *HO* promoter regulation.

### Ash1 provides a mechanism for differential expression between mother and daughter cells via recruitment of Tup1 and Rpd3

Sites of Ash1/Tup1/Rpd3 association tend to be located within large intergenic regions (S6 Fig), suggesting they contribute to regulation of some of the more complex yeast promoters. Ash1 appears to be one of multiple contributors to Tup1 and Rpd3-mediated repression, as loss of Ash1 often caused only slight to moderate reductions in Tup1 and Rpd3 association with the promoters we tested (S4 Table). This data supports previous studies showing that deletions of individual recruiters do not change the genome-wide Tup1 binding pattern, and the number of recruiter binding sites at a given location correlates with the occupancy of Tup1 [42]. Tup1 and Rpd3-regulated genes may therefore have the capacity to respond to multiple different pathways, with each repressor directing association of Tup1 and/or Rpd3 under a unique set of conditions. Many Tup1-Cyc8 recruiters respond to environmental signals; others limit Tup1 repression to a particular cell type. Because Ash1 protein is present predominantly in daughter cells, it is predicted to have much less of a repressive effect in mother cells; thus, Ash1 contributes a unique cell-type specific mode of Tup1 and/or Rpd3 action.

We identified the ORFs downstream of sites of Ash1 localization, for which Ash1 could play a regulatory role (S5 Table). For some of these genes, we can speculate how a repressor localized predominantly in daughter cells might be important, though for many genes it is not clear how a mother-daughter distinction would be advantageous. Ash1 repression of genes encoding cell wall and cell surface proteins, some of which are involved in budding and cytokinesis (S5 Table), could contribute to the polarity that is established between mother and daughter cells. Promoters of some cell cycle regulators also have Ash1 bound (S5 Table). Daughter cells progress through the cell cycle at a different rate than mother cells. Reduced expression of these possible Ash1 target genes, such as *CLN2* and *CDC6*, could contribute to the cell cycle delay in daughter cells. Ash1 may also affect transcription of genes indirectly by tailoring the level of expression of their transcription factors in mother versus daughter cells. Multiple genes encoding DNA-binding factors have Ash1 localized to their upstream region, including several that recruit Tup1-Cyc8 (S5 Table). In this way, Ash1 could indirectly influence the relative expression levels in mothers and daughters for a large number of genes.

## Ash1's recruitment of both Rpd3 and Tup1 may explain its broad spatial and temporal effect on *HO* transcription

The *HO* promoter appears to have characteristics that are not exhibited by the majority of other locations of ATR binding. First, *HO* is the only gene downstream of an ATR peak that is known to be expressed exclusively in mother cells. Mother-specific expression of the Ho endonuclease is critical to ensure that only one cell switches mating type, allowing efficient production of a diploid from a germinating spore. For most Ash1-regulated genes, it is likely that a higher level of expression in mother cells than in daughter cells, without expression being completely "off" in daughters, is advantageous for growth. Second, Tup1 binding to the *HO* promoter is strongly Ash1-dependent (Fig 1), while this is not true for most genes bound by Ash1 and Tup1 (S4 Table). This suggests that Tup1, along with Rpd3, is a necessary component of strong repression of *HO* in daughter cells. Genes that also need to respond to environmental conditions necessitate the use of additional DNA-binding repressors, leading to the observed redundancy of DNA-binding factors that recruit Tup1 and Rpd3. Third, *HO* has two peaks of Ash1 binding, both of which are necessary for obtaining the appropriate level of *HO* expression (Figs 1A and 5). The reason for both peaks is not clear, but could involve limiting the bidirectional nucleosome eviction from the Swi5 sites [62]. Nine other sites throughout the genome share this feature of two peaks, some of which are located between parallel ORFs, upstream of a single gene and/or approximately one kb or less apart, similar to *HO* (listed as "Double" in S3 Table).

The fourth and final feature that distinguishes *HO* from most other sites of association of Ash1, Tup1 and Rpd3 is the observation that the ATR binding sites are concealed by nucleosomes for much of the cell cycle. The majority of ATR sites are depleted of nucleosomes, suggesting Ash1 is similar to many transcription factors, which preferentially bind within NDRs as opposed to binding sites positioned within nucleosomes [66]. Ash1/Tup1 binding to the *HO* promoter was substantially diminished under conditions in which the nucleosomes covering the association sites could not be evicted, and binding was restored when nucleosomes were depleted in the absence of the normally required activators and coactivators (Fig 7). This suggests other Ash1 sites that are concealed likely require dynamic modification or removal of the covering nucleosome at a particular time point to allow Ash1 binding and subsequent recruitment of Tup1 and/or Rpd3. Aside from *HO*, the mechanisms of activation of these promoters and their associated factors are largely unknown. Investigation of the conditions in which these sites are revealed could provide further insight concerning the interplay between chromatin and Ash1 repression.

## The timing of nucleosome eviction may be important for asymmetric *HO* expression

For *HO*, all of these features are likely necessary to constrict expression to a narrow window of the cell cycle, only in mother cells. Expression of the Ho endonuclease outside of this window could be detrimental or counterproductive to the cell due to inappropriate cleavage of DNA. The details concerning the timing of Swi5 binding, exposing of binding sites concealed by nucleosomes, and subsequent Ash1 association, all have important consequences for the asymmetric expression of *HO* in mothers and daughters [4]. Swi5 enters the nucleus as cells enter anaphase and binds to the *HO* promoter. Recruitment of the SWI/SNF chromatin remodeler by Swi5 results in nucleosome eviction and exposure of the Ash1 binding sites, and these events probably occur at the time of cytokinesis. In daughter cells, the large amount of bound Ash1 prevents the promoter from being activatable, presumably because Ash1 recruits Tup1, which blocks coactivator recruitment [41], and the Rpd3 deacetylase complex. In mother cells,

the small amount of Ash1 that binds to the promoter is not sufficient to prevent activation. However, the limited presence of Ash1 does cause mother cell activation of *HO* to be fully dependent on the Gcn5 coactivator [26].

Ash1 is a very unstable protein and is rapidly cleared from the nucleus [67,68]. Experiments show that the effects of Ash1 persist long after the protein is degraded, and at promoter sites far distant from where it binds. In an *ash1* mutant, there is increased association of SWI/SNF, Mediator, and SBF, and evicted nucleosomes are not repopulated within the same time scale as in wild type cells [26,62]. Ash1's ability to recruit both Rpd3 and Tup1, which affect the coactivators and thereby downstream promoter events, likely explains the extent and duration of Ash1's impact. Further studies will be required to fully understand the mechanisms of repression by Ash1.

### A requirement for nucleosome eviction for binding of repressors suggests an interrelationship between activation and repression

Our results demonstrate that the Ash1 repressor requires initial *HO* promoter activation steps for binding. This suggests that achieving appropriate *HO* expression requires not simply a balance of positive and negative transcriptional activities but also a coordination between them. The necessity to restrict *HO* expression to only a few rounds of transcription within a short window of the cell cycle may be the driving factor responsible for integration of activation and repression.

The observation that Ash1 is unable to associate with the *HO* promoter until nucleosomes have been evicted illustrates that dynamic modification of nucleosomes can be required for repression as well as activation. If the mode of Ash1 binding at other intergenic sites concealed by nucleosomes is similar to the *HO* promoter, our data suggests that Ash1 binding to these promoters is also restricted to a short time within the cell cycle or to specific environmental conditions. These genes could therefore represent additional examples of a requirement for activator binding and nucleosome eviction prior to recruitment of repressors and corepressors. Such a scenario may be even more prevalent in higher eukaryotic promoters, some of which require many activating and repressing transcriptional regulators that associate with large enhancer regions [69].

Coordination of positive and negative transcriptional activities could allow a fine tuning of the repression response that may be necessary in cases where the activator is present for a brief period of time or is relatively weak and unable to overcome robust repression already established at the promoter. The repressor would thus temper the coactivator response, and, in a situation such as *HO*, ensure that detrimental levels of transcript are not produced. At regulated promoters, the linkage of activation and repression may also allow activation to trigger a "reset" of the promoter for repression until the next cell cycle. These roles of limiting transcriptional response and resetting the promoter are likely not unique to the Ash1 repressor specifically, as many other proteins that recruit Tup1 and Rpd3 to different sets of genes could perform similar functions. The apparent redundancy of sites of recruitment for Tup1 and Rpd3 to promoters and the ability of some of these sites to be regulated by nucleosome placement thus allows genes to not only respond to different environmental conditions and cellular stresses but also to combine accessible sites and concealed, regulatable sites within the same promoter. These options for building a complex promoter may provide an important level of flexibility in the transcription of highly regulated genes.

## Methods

### Strain construction

All yeast strains used are listed in S1 Table and are isogenic in the W303 background (*leu2-3,112 trp1-1 can1-100 ura3-1 ade2-1 his3-11,15*) [70]. Standard genetic methods were used for

strain construction [71–74]. The *ASH1-V5* C-terminal epitope tag has been described previously [27]. The *TUP1*-V5 and *RPD3*-V5 alleles were constructed as described [73], by integrating a V5 epitope tag with a *HIS3MX* marker from pZC03 (pFA6a-TEV-6xGly-V5-*HIS3MX*), provided by Zaily Connell and Tim Formosa (plasmid #44073; Addgene). For strains with the *HO* -1890 nucleosome replacement, *HO* promoter sequence from -1972 to -1826 was deleted and replaced with *CDC39* ORF sequence from +2583 to +2729, using the *delitto perfetto* method [74]. For strains with the *HO* -1215 nucleosome replacement, *HO* promoter sequence from -1288 to -1139 was deleted and replaced with *CDC39* ORF sequence from +3072 to +3221. Strains with the LexA site upstream of *HIS3* are derived from strain L40 [75]. A plasmid with the *LexA(DBD)-NLS-3xFLAG*::*HphMX* construct was made in several steps (details available on request), and was used to tag the C-terminus of the chromosomal *ASH1* gene [73]. Strains labeled as "Swi5 Site Mut" have an *HO* promoter sequence with mutations of both Swi5 binding sites A and B (a3 and b3 mutations) [63]. For the strain labeled as "Nucl Δ", *HO* sequences from -1268 to -1262 and from -1194 to -1189 were replaced with Reb1 binding sites (TTACCC), which lead to nucleosome depletion [65].

## RNA expression and Chromatin Immunoprecipitation (ChIP) analysis

For logarithmic cell collection ($OD_{660}$ of 0.6 to 0.8), cells were grown at 30˚C in YPA medium (1% yeast extract, 2% bactopeptone, 0.002% adenine) supplemented with 2% dextrose [72]. Cell cycle synchronization was performed by galactose withdrawal and readdition with a *GALp*::*CDC20* strain grown at 25˚C in YPA medium containing 2% galactose and 2% raffinose [21]. Synchrony was confirmed by microscopic analysis of budding indices and analysis of cell-cycle regulated mRNAs.

RNA was isolated from either logarithmically growing cells or synchronized cells, and *HO* mRNA levels were measured by reverse transcription quantitative PCR (RT-qPCR), as described previously [76]. *HO* RNA expression was normalized to that of *RPR1*. *RPR1* encodes the RNA component of RNase P and is transcribed by RNA polymerase III. Most genetic manipulations that affect RNA Pol II transcription do not affect transcription of *RPR1*. For logarithmic cells, normalized *HO* RNA expression values were graphed relative to wild type (WT) expression.

ChIPs were performed as described [21,76], using mouse monoclonal antibodies to the V5 epitope (SV5-Pk1; Abcam) or the FLAG epitope (M2; Sigma) and antibody-coated magnetic beads (Pan Mouse IgG beads; Life Technologies). Cells from either logarithmically growing cells or synchronized cells were cross-linked in 1% formaldehyde for 20 min at room temperature (Ash1, Swi5) or overnight at 4˚C (Tup1) and quenched with 125 mM glycine. ChIP signals were calculated as detailed in the Figure Legends. For some experiments, the concentration of ChIP DNA at the relevant target gene was normalized simply to its corresponding Input DNA and also to a "No Tag" control. For others, samples were first normalized to either an expected negative reference control (IGR-I intergenic region of chromosome I and IGR-V intergenic region of chromosome V) or a known positive reference control (*CLN3* for Ash1, *TEC1* for Tup1, *INO1* for Ume6). For figures using a negative reference control, values were graphed relative to the No Tag control. For figures using a positive reference control, values were graphed relative to the wild type control.

Quantitative PCR (qPCR) experiments for both RNA and ChIP analysis were run on a Roche Lightcycler 480 or a ThermoFisher QuantStudio 3, and concentrations were determined using wild type cDNA or ChIP input for in-run standard curves via the E-method [77]. Error bars represent the standard deviation of at least three biological samples. The Student's t-test was used to determine significance of changes in *HO* expression and factor binding between

different genotypes. For all comparisons mentioned in the Results and Discussion, *p*-values are indicated in the figures. For ChIP tiling PCR across the *HO* promoter (Fig 1A and 1C) and time course experiments, a single sample is shown for simplicity (Figs 1D and 7). Triplicate biological samples for the time course ChIPs in Fig 1D are shown in S1 Fig. Fig 7 contains a single sample for Swi5-V5, Ash1-V5 ChIP and *HO* mRNA, all of which have been confirmed via numerous previous experiments [6,7,27,76].

S2 Table lists the primers used for ChIP and RT-qPCR analysis.

### ChIP-Seq and genomic data analysis

Chromatin isolated from individual, independently collected Ash1-V5, Tup1-V5 or Rpd3-V5 cell pellets was used for multiple ChIPs, performed as described above, which were then pooled for each replicate. Libraries were prepared for triplicate ChIP samples and a single input sample for each strain using the New England Biolabs NEBNext ChIP-Seq Library Prep Reagent Set with dual index primers. Sequencing was performed with an Illumina NovaSeq 6000, 150-bp paired end run (University of Utah High Throughput Genomics Facility). Fastq files were aligned to the genome (UCSC sacCer3) using Novocraft Novoalign version 3.8.1 [78], giving primer adapters for trimming, and allowing for 1 random, multi-hit alignment. Between 10–20 million fragments were mapped with an alignment rate of 98.4–99.7%, and a Pearson correlation >0.9 between replicates based on genomic coverage.

Samples were then processed with MultiRepMacsChIPSeq pipeline version 8 [79]. Alignments over mitochondrial, 2-micron, rDNA, and telomeric regions were discarded from analysis. Excessive duplicate alignments (36–56%) were randomly subsampled to a uniform 20% for each sample. Replicates were depth-normalized, averaged together, and peak calls generated with a minimum size of 200 bp, gap size of 100 bp, and minimum q-value statistic of 2. Peaks were further filtered using the peak score (sum of q-value statistic) using a minimum cutoff of 100. Peaks were annotated by intersection using bedtools [80] with interval files of either genes or intergenic regions.

Data for heat map analysis was collected with BioToolBox get_relative_data with the peak summit using the generated Log2 Fold Enrichment and nucleosome coverage bigWig files, in 25 windows of 20 bp flanking the summit. Heat maps were generated using pHeatmap [81] in custom R scripts.

To determine the position of genome-wide nucleosomes, depth-normalized (Reads Per Million) nucleosomal coverage representing the middle 50% of nucleosomal fragments was generated from [5] using BioToolBox bam2wig version 1.67 [82] by shifting the alignment start position by 37 bp and extending coverage for 76 bp. Mapped nucleosome calls were made with the BioToolBox-Nucleosome version 1 [83] package, map_nucleosomes script with a threshold of 2. Nucleosome calls were filtered with the verify_nucleosome_mapping script using maximum overlap of 35 bp and recenter option. This identified 61,802 nucleosomes. Nucleosomal Depleted Regions were generated as the reciprocal of called nucleosomes using bedtools [80] complement function, which were then filtered for length (75–600 bp) and low residual nucleosome coverage (mean RPM coverage < 2). Nucleosomal edges were generated as intervals 25 bp internal and 10 bp external to the edge coordinates of called nucleosome intervals. ChIP peaks were intersected with nucleosome and NDR intervals using bedtools.

Motif analysis of Ash1 peaks was performed using a 100 bp sequence interval (±50 bp from the called summit of the peak). Motifs displayed in S7 Fig were identified using the MEME-suite [55], with a first order background model. Additional motif analysis was performed with Homer software version 4.10.1 [56,84], using intergenic intervals as a custom background file. Additional searches were performed using only ATR peaks or ATR peaks found in NDRs.

The data used to generate each figure is contained in S6 Table.

## Supporting information

**S1 Appendix. Identification of ORFs downstream of ATR peaks.** This appendix provides information on genes downstream of ATR peaks, that are potentially regulated by Ash1, Tup1, and Rpd3.
(DOCX)

**S2 Appendix. Attempts to identify Ash1 binding site locations within the *HO* promoter.** This appendix describes experiments where potential Ash1 binding sites within the *HO* promoter were mutated and effects on *HO* expression were determined.
(DOCX)

**S1 Fig. Ash1 facilitates Tup1 recruitment to the *HO* promoter.** Data from Fig 1D is shown along with two additional replicates of the experiment. Binding of Tup1-V5 was measured by ChIP analysis with cells containing the *GALp::CDC20* allele and synchronized by galactose withdrawal and readdition. The 0 min time point represents the G2/M arrest, before release with galactose addition. Cells were harvested at the indicated time points following release (x-axis), and samples were processed for ChIP analysis. Graphs show binding of Tup1-V5 in wild type (blue) and *ash1* (green) cells, at the *HO* Upstream Site (left) and *HO* Downstream Site (right). Enrichment for each sample at *HO* was normalized to enrichment at an intergenic region on chromosome I (IGR-I) and to the corresponding input sample.
(TIF)

**S2 Fig. A multicopy *ASH1* plasmid increases *ASH1* mRNA and decreases *HO* mRNA levels.** (A) A YEp-*ASH1* multicopy plasmid results in increased *ASH1* mRNA. *ASH1* mRNA analysis under conditions of *ASH1* overexpression, using cell samples identical to those in Fig 1E (Tup1-V5 ChIP analysis). Strains were transformed with a pRS426 YEp-*URA3* vector, either empty (blue) or containing *ASH1* (green). *ASH1* mRNA levels were measured, normalized to *RPR1*, and expressed relative to wild type. Each dot represents a single data point, and error bars reflect the standard deviation. (B) A YEp-*ASH1* multicopy plasmid results in decreased *HO* mRNA levels. *HO* mRNA analysis under conditions of *ASH1* overexpression, using cell samples identical to those in Fig 1E (Tup1-V5 ChIP analysis). Strains were transformed with a pRS426 YEp-*URA3* vector, either empty (blue) or containing *ASH1* (green). *HO* mRNA levels were measured, normalized to *RPR1*, and expressed relative to wild type. Each dot represents a single data point, and error bars reflect the standard deviation.
(TIF)

**S3 Fig. Correlation between targeted ChIP and ChIP-Seq.** Correlation plots showing Ash1-V5 (A), Tup1-V5 (B) and Rpd3-V5 (C) $\log_2$ fold enrichment signals obtained via traditional ChIP (y-axis) and ChIP-Seq (x-axis). The genes tested are detailed in S4 Table. Gene common names identify some of the dots in the plots, including the *HO* Downstream site, *CLN3* (used as positive control for Ash1-V5 ChIPs), *TEC1* (used as positive control for Tup1-V5 ChIPs; very low Ash1-V5 binding), *INO1* (used a positive control for Rpd3-V5 ChIPs; not bound by Ash1-V5), and *POG1* (a high-scoring Ash1-V5 peak that shows co-localization with Tup1-V5 and Rpd3-V5). The $R^2$ value obtained from linear regression analysis of each plot is shown.
(TIF)

**S4 Fig. Browser snapshots to display overlap of Ash1, Tup1 and Rpd3.** Additional snapshots of ChIP-Seq results from the Genome Browser IGV (Broad Institute), showing sequenced

fragment pileups for the portion of the indicated chromosome, autoscaled for each factor independently (Refer to Fig 4B for another snapshot). The top track (gray) for each set shows MNase-Seq for nucleosome positioning reference. The colored tracks show ChIP-Seq results for Ash1-V5 (red), Tup1-V5 (blue) and Rpd3-V5 (green). The bottom track displays gene annotation. Gene names are indicated only for those with start sites downstream of a site of Ash1-V5, Tup1-V5, and Rpd3-V5 co-enrichment.
(TIF)

**S5 Fig. Tup1 and Rpd3 show substantial overlap at many genomic locations.** Heat maps depict the $\log_2$ fold enrichment of Ash1-V5, Tup1-V5 and Rpd3-V5 from -500 to +500 nucleotides relative to the center of each reference peak, in bins of 20-bp. The color scale at the right indicates the level of $\log_2$ fold enrichment for each factor. Each horizontal line depicts a single peak of enrichment. (A) Tup1 peaks (816) used as the reference. (B) Rpd3 peaks (1343) used as the reference.
(TIF)

**S6 Fig. ATR peaks are preferentially located in very large intergenic regions.** Shown is the percent of intergenic regions (y-axis) within each of six size categories of intergenic regions (x-axis). Distribution of genome-wide intergenic regions is shown in blue, and distribution of intergenic regions containing ATR co-localized peaks is shown in red.
(TIF)

**S7 Fig. Motifs identified from MEME analysis of Ash1 peaks.** The top two motifs identified from MEME analysis of Ash1 peaks are shown. Motif 1 is found in 68 of the 250 Ash1 peaks, and Motif 2 was identified in 49 Ash1 peaks. Motif 2 resembles an Mcm1 motif [58,59]. The *HO* sequence from -1244 to -1229 is shown below Motif 2, to which it bears some similarity. Combined mutation of all positions in this region of the *HO* promoter (underlined) only modestly decreased Ash1 binding (S2 Appendix).
(TIF)

**S8 Fig. Browser snapshots of three types of ATR peaks.** IGV genome browser snapshots of sequenced fragment pileups are shown to demonstrate the *HO* promoter (A) and two examples of ATR peaks from each category in Table 3 (B; NDR, NDR/Nucleosome Boundary and Nucleosome). Each factor was autoscaled independently. Tracks include: MNase-Seq nucleosome positions (gray), fragment density of Ash1-V5 (red), Tup1-V5 (blue) and Rpd3-V5 (green), annotations of peaks (beneath each fragment density track), gene annotation, position of the Ash1 peak summit, and mapped NDRs and nucleosomes (using the MNase-Seq data).
(TIF)

**S1 Table. Strains used in this study.**
(DOCX)

**S2 Table. Primers for ChIP and RT-qPCR analysis.**
(DOCX)

**S3 Table. Information on each Ash1 peak from the ChIP-Seq analysis.** For each Ash1 peak, the chromosomal location is given along with log2 fold enrichment. Each peak is identified as ATR (Ash1-Tup1-Rpd3), AT (Ash1-Tup1), AR (Ash1—Rpd3), or A-only (Ash1).
(XLSX)

**S4 Table. Analysis of Ash1, Tup1, and Rpd3 binding to a subset of diverse promoters.**
(DOCX)

**S5 Table. Information on ORFs located downstream of ATR.**
(XLSX)

**S6 Table. Values for data used to create graphs in the figures.** The Excel file contains multiple tabs, with each tab containing the data for a single figure.
(XLSX)

## Acknowledgments

We thank Tim Formosa and Bobby Yarrington for advice throughout the course of these experiments and Tim Formosa and Dean Tantin for comments on the manuscript. We thank Zaily Connell, Tim Formosa, Mark Hochstrasser, Anita Sil, and Warren Voth for providing plasmids used in strain construction. We also thank Frank Pugh for communicating information and David Virshup for a yeast strain.

## Author Contributions

**Conceptualization:** Emily J. Parnell, David J. Stillman.

**Data curation:** Emily J. Parnell, Timothy J. Parnell, Chao Yan.

**Formal analysis:** Emily J. Parnell, Timothy J. Parnell, Chao Yan.

**Funding acquisition:** Lu Bai, David J. Stillman.

**Investigation:** Emily J. Parnell, Chao Yan, David J. Stillman.

**Methodology:** Emily J. Parnell, Timothy J. Parnell, Lu Bai, David J. Stillman.

**Project administration:** Emily J. Parnell, Lu Bai, David J. Stillman.

**Resources:** Lu Bai, David J. Stillman.

**Software:** Timothy J. Parnell.

**Supervision:** Lu Bai, David J. Stillman.

**Validation:** Emily J. Parnell.

**Visualization:** Emily J. Parnell, Timothy J. Parnell, Lu Bai, David J. Stillman.

**Writing – original draft:** Emily J. Parnell.

**Writing – review & editing:** Timothy J. Parnell, Lu Bai, David J. Stillman.

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
