## [Decision Letter · Decision Letter 0]

27 Oct 2020

Dear David,

Thank you very much for submitting your Research Article entitled 'Ash1 and Tup1 Dependent Repression of the Saccharomyces cerevisiae HO promoter Requires Activator-Dependent Nucleosome Eviction' to PLOS Genetics. Your manuscript was fully evaluated at the editorial level and by independent peer reviewers. The reviewers appreciated the attention to an important topic but identified some aspects of the manuscript that should be improved. In particular, reviewer 2 raised the valid point that the nucleosome mapping experiment was carried out with log phase cells, which is not the ideal population for studying expression of HO. However, it would be extremely difficult to map transient nucleosomes, and this is probably best dealt with by discussing the limitations of the analysis. There are a few other points relating to interpretation and suggested controls that should be relatively easy to address.

We therefore ask you to modify the manuscript according to the review recommendations before we can consider your manuscript for acceptance. Your revisions should address the specific points made by each reviewer.

[LINK]

Yours sincerely,

Geraldine Butler

Associate Editor

PLOS Genetics

Gregory P. Copenhaver

Editor-in-Chief

PLOS Genetics

Reviewer's Responses to Questions

**Comments to the Authors:**

Reviewer #1: In this manuscript, Parnell et al. continue the analysis of HO gene regulation in yeast that has been a long-term project of the Stillman lab. The current work focuses on the association of the Ash1 repressor, which is required for repression of HO in daughter cells following cell division, with the HO promoter. Using standard ChIP and ChIP-seq, they identify two regions in the HO promoter with which Ash1 associates, as well as >200 genomic sites. Ash1 co-localizes with the co-repressors Tup1 and Rpd3 at HO and the majority of its other genomic sites, and contributes to their binding in most cases, though the mechanism of binding of Ash1 remains unclear. At HO, Ash1 binds to sites that are nucleosomal through most of the cell cycle, excepting for the window following Swi5 binding, and Swi5 is required for Ash1 binding. In addition, depleting the nucleosomes at which Ash1 binds by inserting Reb1 sites allows Swi5-independent Ash1 association.

Technically the work is solid, and the authors are clear in explaining their use of replicates and when they are not needed (Figure 7A, which corroborates much previous work). The results have novel aspects and provide new insight into mechanisms of cell cycle regulation and repression via Ash1, as well as identifying numerous targets of this repressor about which little is presently known. As to broader significance, based on trends in citations, publications, and grant awards (generally, not specific to these authors), I think the issues addressed in this work do not hold as much general interest as they once did. Nonetheless I would favor publication, as the work adds significantly to the impressive knowledge of the complex regulation of this paradigmatic promoter and should be of interest to those studying mechanisms by which chromatin regulates transcription.

Some minor points:

1) Results for an untagged control are presented for standard ChIP but apparently this control was not performed for ChIP-seq experiments. As I imagine the ChIP-seq experiments were performed in Dr. Bai’s lab, I wonder if she has results from any untagged control using the anti-V5 (or other) antibodies that could be shown together with the heat maps in Figure 6? I think it is not likely the results will change but it would be a nice control if available.

2) State in Figure 1 legend that replicates are shown in Fig S1.

3) Table S5: It would be helpful to provide p-values for enrichment for the various categories shown. What is the source of information for stress responsive genes and for the other categories? I assume they correspond to standard Gene Ontology categories?

4) Line 598, “switch” should be “switches”

5) Ref 62 is incomplete—in press?

6) Methods—presumably perfetto delitto was used for nucleosome replacements? This (or whatever method was used) should be stated.

Reviewer #2: In this study, the authors investigate the role of the Ash1 transcription factor in regulation of the budding yeast HO promoter. The HO gene is an important model gene because of its very long promoter (~2.5 kb), which is much more typical of the genes of higher eukaryotes than of other yeast genes. The Stillman Lab has studied the cell cycle dependence of HO regulation in great depth; here they provide additional insight into its control. HO is expressed only in mother cells. Ash1 is a negative transcription factor that accumulates in daughter cells and is primarily responsible for mother cell-specific HO expression. The authors show using ChIP that Ash1 binds to two sites in the HO promoter that appear to be located inside nucleosomes; these sites are atypical in that most Ash1 binding sites are located in nucleosome-depleted regions (NDRs). They go on to show that Ash1 recruits both the Tup1 co-repressor and the Rpd3 histone deacetylase complex to the HO promoter. Finally, Ash1 binding at HO is shown to be dependent on the Swi5 transcription activator. Interestingly, if the nucleosomal sequence where Ash1 binds is mutated to include a pair of Reb1 binding sites, creating an NDR, Ash1 binding becomes Swi5-independent, suggesting that this nucleosome is a direct regulator of Ash1 binding. The authors propose that nucleosomes block Ash1 binding at HO and must be removed in a Swi5-dependent process before Ash1 can bind, thus ensuring appropriate timing of Ash1 binding during the cell cycle.

The manuscript is interesting, represents an important advance in our understanding of HO regulation and is clearly written. I have the following comments:

1. The nucleosome positions on the HO promoter are derived from MNase-seq data from another study, presented in Fig. 1A. They are critical for the interpretation of the data, but the nucleosome positions were determined using log-phase cells and are therefore averaged over the cell cycle. Since Ash1 binding is cell cycle-dependent, it is possible that the nucleosome positions on the HO promoter change during the cell cycle and that their average positions are misleading. In addition, there is the complication that Ash1 may bind only in daughter cells and that the chromatin structures of mother and daughter cells also differ. The question that arises is whether the Ash1 binding site is always in a nucleosome, except when activated, as proposed? If Ash1 binds at HO for only ~10 minutes of the cell cycle and only in daughter cells, then a transient NDR where Ash1 binds might be expected in only perhaps 5% of cells in a log-phase population, which would be very difficult to detect with any certainty in MNase-seq data.

2. An associated point is that the authors represent the nucleosomes as if they cover only 76 bp. This is often done to make the nucleosome map easier to interpret. However, it is a mis-representation of the data since the nucleosome covers 147 bp.

3. Page 9: "An rpd3 null single mutant did not change expression of HO in the bulk population, but single-cell analysis demonstrated that HO was expressed in approximately 50% of the daughter cells." Could this be explained if the cells are producing the same amount of HO mRNA but, in rpd3-null cells, it is not being correctly partitioned between mother and daughter?

4. The ChIP-seq data for Ash1, Tup1 and Rpd3 appear to be of excellent quality (Figs. 4B and S4). The authors should show the ChIP-seq data for the HO gene.

5. The ChIP data as a function of the cell cycle show that the Swi5 activator binds first, then the Ash1 repressor (recruiting Tup1) and then HO transcription begins (Fig. 7A). Presumably, Ash1 is only binding and repressing in the daughter cells, while mother cells continue with activation. I think this could be stated more clearly in the Discussion.

Minor points:

6. The ChIP-seq data have been submitted to GEO but can't be accessed by the reviewers. I think that appropriate biological replicate experiments have been done, but I can't confirm that this is the case.

7. Figure 6: It is unclear what is plotted in the heat maps (define nucleosome density).

**Have all data underlying the figures and results presented in the manuscript been provided?**

Reviewer #1: **No: **Data for qPCR analysis of standard ChIP experiments is missing

Reviewer #2: Yes

PLOS authors have the option to publish the peer review history of their article (what does this mean?). If published, this will include your full peer review and any attached files.

Reviewer #1: No

Reviewer #2: No

---

## [Editor Report · Decision Letter 1]

25 Nov 2020

Dear Dr Stillman,

We are pleased to inform you that your manuscript entitled "Ash1 and Tup1 Dependent Repression of the Saccharomyces cerevisiae HO promoter Requires Activator-Dependent Nucleosome Eviction" has been editorially accepted for publication in PLOS Genetics. Congratulations!

Yours sincerely,

Geraldine Butler

Associate Editor

PLOS Genetics

Gregory P. Copenhaver

Editor-in-Chief

PLOS Genetics

Comments from the reviewers (if applicable):

**Data Deposition**

http://datadryad.org/submit?journalID=pgenetics&manu=PGENETICS-D-20-01462R1

**Press Queries**

---

## [Editor Report · Acceptance letter]

23 Dec 2020

PGENETICS-D-20-01462R1 

Ash1 and Tup1 Dependent Repression of the Saccharomyces cerevisiae * HO * promoter Requires Activator-Dependent Nucleosome Eviction 

Dear Dr Stillman, 

We are pleased to inform you that your manuscript entitled "Ash1 and Tup1 Dependent Repression of the Saccharomyces cerevisiae * HO * promoter Requires Activator-Dependent Nucleosome Eviction" has been formally accepted for publication in PLOS Genetics! Your manuscript is now with our production department and you will be notified of the publication date in due course.

With kind regards,

Melanie Wincott

PLOS Genetics

On behalf of:
